# Improving Tabular Generative Models: Loss Functions, Benchmarks, and Iterative Objective Bayesian Approaches

## Abstract

Access to extensive data is essential for improving model performance and generalization in deep learning (DL). When dealing with sparse datasets, a promising solution is to generate synthetic data using deep generative models (DGMs). However, these models often struggle to capture the complexities of real-world tabular data, including diverse variable types, imbalances, and intricate dependencies. Additionally, standard Bayesian optimization (SBO), commonly used for hyperparameter tuning, struggles with aggregating metrics of different units, leading to unreliable averaging and suboptimal decisions.

To address these gaps, we introduce a novel correlation- and distribution-aware loss function that regularizes DGMs, enhancing their ability to generate synthetic tabular data that faithfully represents actual distributions. To aid in evaluating this loss function, we also propose a new multi-objective aggregation method using iterative objective refinement Bayesian optimization (IORBO) and a comprehensive statistical testing framework. While the focus of this paper is on improving the loss function, each contribution stands on its own and can be applied to other DGMs, applications, and hyperparameter optimization techniques.

We validate our approach using a benchmarking framework with twenty real-world datasets and ten established tabular DGM baselines. Results demonstrate that the proposed loss function significantly improves the fidelity of the synthetic data generated with DGMs, leading to better performance in downstream machine learning (ML) tasks. Furthermore, the IORBO consistently outperformed SBO, yielding superior optimization results. This work advances synthetic data generation and optimization techniques, enabling more robust applications in DL.

## 1 Introduction

For a wide range of deep learning (DL) applications, additional data is crucial for improving both model performance and generalization. The fast-paced advancements in deep generative modeling have opened exciting possibilities for data synthesis. Models trained on images and text (Karras et al., 2021; Team et al., 2023) effectively learn probability distributions over complex data and generate high-quality, realistic samples. This success on structured data has fueled a surge in deep generative model (DGM)-based methods (Goodfellow et al., 2014) for tabular data generation in recent years. However, modeling tabular data presents unique challenges due to the lack of clear structure and the presence of both continuous and discrete variables with complex interactions, imbalances, and non-linear relationships.

Existing deep neural network (DNN) models often fail to effectively capture the complexities in tabular data, struggling to approximate even basic statistics such as the mean and variance of a variable (Xu et al., 2019). Moreover, tabular data inherently contains structure and correlations that DNNs find particularly challenging to learn. Current approaches to improve downstream machine learning (ML) analyses focus primarily on addressing data imbalance (Xu et al., 2019; Sun et al., 2023; Zhao et al., 2021), while neglecting the equally crucial roles of feature distribution and correlation analysis. To overcome this gap, we propose a novel correlation- and distribution-aware loss function for DGMs, designed to enforce these statistics in generative models. This loss function works with various DGMs and promotes more effective modeling of the complex relationships

within tabular data. To address the growing use of DGMs for tabular data, we also introduce a benchmarking framework based on statistical tests.

Hyper-parameter search is essential for optimizing DGMs performance, and Bayesian optimization (BO) efficiently fine-tunes these parameters to improve outcomes without exhaustive trials. However, Standard Bayesian optimization (SBO) struggles to aggregate multiple metrics with different units, making mean aggregation unreliable and leading to sub-optimal decisions. To overcome this limitation, we introduce iterative objective refinement Bayesian optimization (IORBO), which aggregates metrics by ranks, enabling meaningful comparisons across diverse objectives and paving the way for more robust optimization strategies.

This work focuses on enhancing the performance of DGMs through a novel loss function, supported by a new multi-objective aggregation method and a comprehensive statistical testing framework that strengthen the performance and evaluation of our approach. In summary, we provide:

**(1) A Correlation- and Distribution-Aware Loss Function:** We propose a custom correlation- and distribution-aware loss function that emphasizes the importance of feature correlation and distribution in tabular data. Acting as a regularizer, this custom loss function significantly enhancing the performance of DGMs, including generative adversarial network (GAN), variational auto-encoder (VAE), and denoising diffusion probabilistic model (DDPM), as demonstrated through extensive benchmark evaluations.

**(2) Benchmarking Framework for Synthetic Data Generation Algorithms:** We establish a comprehensive open-source benchmarking framework that includes twenty tabular datasets and various evaluation metrics based on statistical tests. This framework implements ten state-of-the-art tabular DGMs and supports extensions with additional methods and datasets.

**(3) Iterative Objective Refinement Bayesian Optimization:** We propose IORBO to aggregate multiple objectives through ranking, resolving inconsistencies caused by metrics with different units or scales.

## 2 RELATED WORK

Most existing methods to generate synthetic tabular data developed in the past decade model measurements in a table as a joint parametric density and then sample from that parametric model. Different models have been employed based on data characteristics: multivariate Gaussian (Frühwirth-Schnatter et al., 2018), Bayesian networks (Aviñó et al., 2018; Zhang et al., 2017), and copulas (Patki et al., 2016) for non-linearly correlated continuous variables. However, these methods are limited by their inability to capture complex relationships beyond the chosen model types.

The remarkable performance and flexibility of DGMs, particularly VAEs (Kingma and Welling, 2013), diffusion models (Sohl-Dickstein et al., 2015; Kotelnikov et al., 2023), and GANs with their numerous extensions (Arjovsky et al., 2017; Gulrajani et al., 2017; Zhu et al., 2017; Yu et al., 2017), have made them very appealing for data representation. This appeal extends to generating tabular data, especially in the healthcare domain. For example, Yahi et al. (2017) leveraged GANs to create synthetic continuous time-series medical records, and Camino et al. (2018) proposed to generate discrete tabular healthcare data using GANs. `CTGAN` (Xu et al., 2019), `DP-CGANS` (Sun et al., 2023) and `CTAB-GAN` (Zhao et al., 2021) were proposed to address the complexities of mixed-type tabular data and to address challenges when generating realistic synthetic data, particularly for imbalanced datasets. `TabDDPM` (Kotelnikov et al., 2023) is a diffusion model designed specifically for tabular data offering the flexibility to incorporate various backbone architectures to model the reverse process.

## 3 METHODS

DGMs learn to map a random noise vector, denoted by $z$, to an output sample. This allows them to generate new data instances that resemble the training data. DGMs have found various applications, *e.g.*, to generate images (Goodfellow et al., 2014; Karras et al., 2020), multi-modal medical images (Zhu et al., 2017), or vectors of tabular data (Xu et al., 2019; Sun et al., 2023; Zhao et al., 2021). In this work, the focus was to generate tabular data with continuous and discrete variables.

### 3.1 A Correlation- and Distribution-Aware Loss Function

Let the training dataset be $\mathbf{X} = \{\boldsymbol{x}_i = (\boldsymbol{x}_i^{(c)}, \boldsymbol{x}_i^{(d)}) : \forall i \in \{1, \ldots, N\}\}$, where $N$ is the number of training samples. The $\boldsymbol{x}_i \in \mathbb{R}^m$ denotes the $i$-th training sample from $\mathbf{X}$, and $\boldsymbol{x}_i^{(c)}$ and $\boldsymbol{x}_i^{(d)}$ are continuous and discrete features, respectively. Let $p_{\tilde{\mathbf{X}}}$ be the learned probability density over the synthetic data, $\tilde{\boldsymbol{x}}$, such that $\tilde{\boldsymbol{x}} \in \mathbb{R}^m$ is a sample from the DGM, $\mathcal{G}$. Here, $\mathcal{G}$ is a learned mapping from a prior distribution $p(\boldsymbol{z})$ to the data space $p(\boldsymbol{x}|\boldsymbol{z})$.

*Correlation-aware loss function.* The correlation-aware loss function is defined as

$$\mathcal{L}_{\text{correlation}} = \frac{2}{m(m-1)} \sum_{j=1}^{m} \sum_{k=j+1}^{m} (\boldsymbol{g}_{j,k} - \tilde{\boldsymbol{g}}_{j,k})^2, \tag{1}$$

where $\boldsymbol{g}$ is the sample correlation over the real data and $\tilde{\boldsymbol{g}}$ is the sample correlation over the generated data, such that

$$\boldsymbol{g}_{j,k} = \frac{1}{N} \sum_{i=1}^{N} \frac{\boldsymbol{x}_{i,j} - \boldsymbol{\mu}_j}{\boldsymbol{\sigma}_j + \epsilon} \cdot \frac{\boldsymbol{x}_{i,k} - \boldsymbol{\mu}_k}{\boldsymbol{\sigma}_k + \epsilon} \qquad \text{and} \qquad \tilde{\boldsymbol{g}}_{j,k} = \frac{1}{B} \sum_{i=1}^{B} \frac{\tilde{\boldsymbol{x}}_{i,j} - \tilde{\boldsymbol{\mu}}_j}{\tilde{\boldsymbol{\sigma}}_j + \epsilon} \cdot \frac{\tilde{\boldsymbol{x}}_{i,k} - \tilde{\boldsymbol{\mu}}_k}{\tilde{\boldsymbol{\sigma}}_k + \epsilon}, \tag{2}$$

with $B$ the size of the mini-batch used when training the DGM, and elements $\boldsymbol{x}_{i,j}$ and $\tilde{\boldsymbol{x}}_{i,j}$ belonging to vectors $\boldsymbol{x}_i \in \mathbf{X}$ and $\tilde{\boldsymbol{x}}_i \in \tilde{\mathbf{X}}$, respectively. A small positive value, $\epsilon = 1 \cdot 10^{-5}$, was added to the denominators of the correlation terms to avoid division by zero. The mean and standard deviation of the $j$-th column in a tabular data set, $\mathbf{X}$, were estimated as

$$\boldsymbol{\mu}_j = \frac{1}{N} \sum_{i=1}^{N} \boldsymbol{x}_{i,j} \qquad \text{and} \qquad \boldsymbol{\sigma}_j = \sqrt{\frac{1}{N} \sum_{i=1}^{N} (\boldsymbol{x}_{i,j} - \boldsymbol{\mu}_j)^2}. \tag{3}$$

Similarly, $\tilde{\boldsymbol{\mu}}_j$ and $\tilde{\boldsymbol{\sigma}}_j$ were estimated as the mean and standard deviation of the generated data, $\{\tilde{\boldsymbol{x}}_i : \forall i \in \{1, \ldots, B\}\}$.

*Distribution-aware loss function.* The distribution-aware loss function integrates the strengths of the method of moments and maximum likelihood estimation (MLE) to align with the true distribution by capturing both statistical moments and likelihood properties in order to enhance the model's ability to learn accurate data representations (Pearson, 1936; Rice, 2007). Additionally, the choice of moments over distance-based metrics, such as Wasserstein, is motivated by their computational efficiency and stability, as lower-order moments provide a robust approximation of the distribution while avoiding the high computational cost associated with distance-based methods. To characterize the training data distribution, we employed the raw first and central second moments,

$$\mathcal{S}_j^{(1)} = \frac{1}{N} \sum_{i=1}^{N} \boldsymbol{x}_{i,j} = \boldsymbol{\mu}_j \qquad \text{and} \qquad \mathcal{S}_j^{(2)} = \frac{1}{N} \sum_{i=1}^{N} (\boldsymbol{x}_{i,j} - \boldsymbol{\mu}_j)^2 = \boldsymbol{\sigma}_j^2, \tag{4}$$

and for $h \geq 3$ the standardized higher moments,

$$\mathcal{S}_j^{(h)} = \frac{1}{N} \sum_{i=1}^{N} \left( \frac{\boldsymbol{x}_{i,j} - \boldsymbol{\mu}_j}{\boldsymbol{\sigma}_j} \right)^h = \boldsymbol{\gamma}^h. \tag{5}$$

Similarly, the empirical moments were computed for the synthetic data, denoted as $\tilde{\mathcal{S}}_j^{(1)}$, $\tilde{\mathcal{S}}_j^{(2)}$, and $\tilde{\mathcal{S}}_j^{(h)}$, again for $h \geq 3$. In this case, $B$ was used in place of $N$. Finally, the distribution loss was defined as

$$\mathcal{L}_{\text{distribution}} = \frac{1}{m} \sum_{j=1}^{m} \sum_{h=1}^{H} \frac{1}{h} \left( 1 - \frac{\tilde{\mathcal{S}}_j^{(h)} + \epsilon}{\mathcal{S}_j^{(h)} + \epsilon} \right)^2 \tag{6}$$

$$= \frac{1}{m} \sum_{j=1}^{m} \left( \left( 1 - \frac{\tilde{\boldsymbol{\mu}}_j + \epsilon}{\boldsymbol{\mu}_j + \epsilon} \right)^2 + \frac{1}{2} \left( 1 - \frac{\tilde{\boldsymbol{\sigma}}_j^2 + \epsilon}{\boldsymbol{\sigma}_j^2 + \epsilon} \right)^2 + \sum_{h=3}^{H} \frac{1}{h} \left( 1 - \frac{\tilde{\boldsymbol{\gamma}}^h + \epsilon}{\boldsymbol{\gamma}^h + \epsilon} \right)^2 \right), \tag{7}$$

where the number of moments, $H$, was hyper-parameter Instead of making the moments equal, their quotient was made to be equal to one as a way to handle scale differences. By using a unified distribution-aware loss, we handle continuous and discrete variables in the same manner, simplifying implementation and preventing imbalances that could arise from separate regularization terms for different data types.

*Custom loss function for DGMs.* The correlation- and distribution-aware loss function was integrated into three prominent DGMs: GAN, VAE, and DDPM. For GANs, the proposed loss function was incorporated into the generator's loss

$$\widetilde{\mathcal{L}}_G = \underbrace{\mathbb{E}_{\boldsymbol{z} \sim p_{\boldsymbol{z}}(\boldsymbol{z})} \big[ \log(1 - D(G(\boldsymbol{z}))) \big]}_{\mathcal{L}_G} + \alpha \mathcal{L}_{\text{correlation}} + \beta \mathcal{L}_{\text{distribution}}, \tag{8}$$

where $\mathcal{L}_G$ is the original GAN's generator loss, and $G$ and $D$ the generator and discriminator of the GAN, respectively. The hyper-parameters, $\alpha$ and $\beta$, controlled the influence of the correlation and distribution terms.

We extended the `TVAE` model (Xu et al., 2019) (a VAE designed for tabular data) with the proposed loss function

$$\widetilde{\mathcal{L}}_{\text{TVAE}} = \underbrace{\mathcal{L}_{\text{reconstruction}} + \mathcal{L}_{\text{KLD}}}_{\mathcal{L}_{\text{TVAE}}} + \alpha \mathcal{L}_{\text{correlation}} + \beta \mathcal{L}_{\text{distribution}}, \tag{9}$$

where $\mathcal{L}_{\text{TVAE}}$ is the original `TVAE`'s loss, and $\mathcal{L}_{\text{reconstruction}}$ and $\mathcal{L}_{\text{KLD}}$ are the reconstruction loss and the Kullback–Leibler (KL) regularization term, respectively.

For the diffusion model, `TabDDPM` (Kotelnikov et al., 2023), the proposed loss function was integrated into the total loss of the multinomial diffusions as

$$\widetilde{\mathcal{L}}_{\text{TabDDPM}} = \underbrace{\mathcal{L}_t^{\text{simple}} + \frac{\sum_{i \leq C} L_t^i}{C}}_{\mathcal{L}_{\text{TabDDPM}}} + \alpha \mathcal{L}_{\text{correlation}}^{(d)} + \beta \mathcal{L}_{\text{distribution}}^{(d)} + \zeta \mathcal{L}_{\text{distribution}}^{(c)}, \tag{10}$$

where $\mathcal{L}_{\text{TabDDPM}}$ denotes the original `TabDDPM` loss, comprising the mean-squared error for the Gaussian diffusion term, $\mathcal{L}_t^{\text{simple}}$, and the KL divergence for all multinomial diffusion terms, $\sum_{i \leq C} L_t^i / C$ (Kotelnikov et al., 2023).

Unlike other DGMs, `TabDDPM` handles continuous and discrete features separately. For continuous features, `TabDDPM` predicts the Gaussian noise added through a forward Markov process. For discrete features, it predicts their one-hot encoded representation. To align our proposed loss functions with this characteristic, we adapted the correlation and distribution loss functions, $\mathcal{L}_{\text{correlation}}^{(d)}$ and $\mathcal{L}_{\text{distribution}}^{(d)}$, to focus exclusively on discrete features. For continuous features, the Gaussian input noise is treated as the real data and the `TabDDPM`'s predicted noise component as the synthetic data, incorporating a controlling parameter $\zeta$ into the $\mathcal{L}_{\text{distribution}}^{(c)}$ computation.

## 3.2 EVALUATION

*Statistical similarity.* The statistical similarity evaluation focuses on how well the statistical properties of the real training data are preserved in the synthetic data. Inspired by a previous review study (Goncalves et al., 2020), we compared two aspects: (1) Individual variable distributions assess how closely the distributions of each variable in the real and synthetic data sets resemble each other; and (2) pairwise correlations reveal the differences in pairwise correlations between variables across the real and synthetic data (Step 1 in Figure 1).

We employed four metrics to quantify how closely the real and synthetic data distributions resemble each other: the KL divergence (Hershey and Olsen, 2007), the Pearson's Chi-Square (CS) test (Pearson, 1992), the Kolmogorov–Smirnov (KS) test (Massey Jr, 1951) and the dimension-wise probability (DWP) (Armanious et al., 2020). To assess how effectively the synthetic data captures the inherent relationships between variables observed in the real data, we used the Pearson correlation coefficient and Cramer's V coefficient (Frey, 2018) (see Section A in the Appendix for more details).

*ML performance.* The ML performance evaluation is meant to enable researchers to leverage synthetic data when developing ML methods in two key areas: Train-Synthetic-Test-Real (TSTR) (Lu

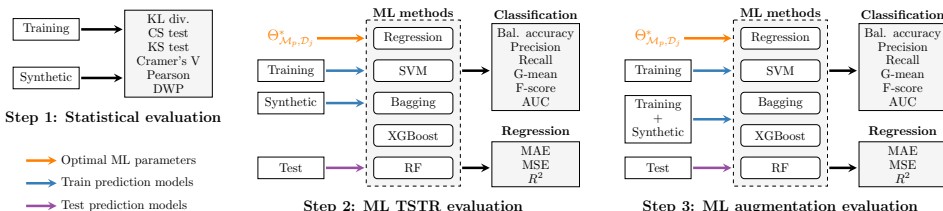

Figure 1: Evaluation pipeline. For dataset $\mathcal{D}_j$ and ML method $\mathcal{M}_p$, the optimal hyper-parameters, $\Theta^*_{\mathcal{M}_p, \mathcal{D}_j}$, were determined using five-fold cross-validation based on ML evaluation metrics (see Figure 4 in the Appendix).

et al., 2023) and augmentation (see Figure 1 and Steps 2 and 3). In the TSTR task (Step 2 in Figure 1), the goal for ML methods trained on synthetic data was to achieve performance comparable or identical to those trained on real data. This work introduces the concept of an ML augmentation task, which, to our knowledge, is the first application of its kind when evaluating tabular synthetic data (Step 3 in Figure 1). Here, the objective was for models trained on a combination of real and synthetic data to outperform models trained solely on real data. By incorporating synthetic data, the models can potentially learn from a richer dataset and achieve improved performance.

To comprehensively evaluate the performance of trained ML models on imbalanced classification datasets, we employed a suite of metrics including balanced accuracy, precision, recall, geometric mean (G-mean), F-score, and area under the ROC curve (AUC). For regression, we used metrics focused on capturing regression error: mean absolute error (MAE), mean squared error (MSE), and the coefficient of determination, $R$-squared ($R^2$). This combined evaluation approach provides a nuanced understanding of model performance across both classification and regression tasks.

## 3.3 HYPER-PARAMETER SEARCH

Hyper-parameters play a pivotal role in tailoring ML methods and DGMs to specific datasets and achieving optimal performance. To systematically optimize the hyper-parameters, we employed BO, a powerful technique to efficiently explore black-box functions. Specifically, we utilized the tree-structured parzen estimator approach (TPE) algorithm (Bergstra et al., 2011) within the `Hyperopt`[1] library to identify optimal hyper-parameter configurations for each combination. This approach enabled us to effectively navigate the complex hyper-parameter space and select the most suitable settings for the experiments.

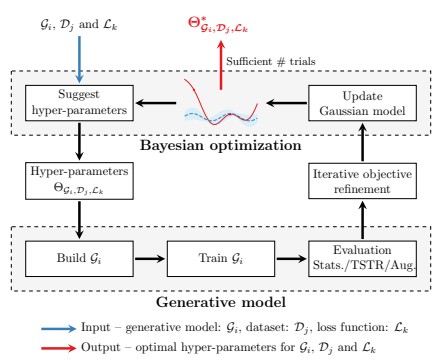

Figure 2: Hyper-parameter search for a single generative model.

We conducted two distinct tuning processes. First, each ML method used in ML TSTR and augmentation evaluation (Figure 1 and Step 2 and 3), was fine-tuned for each dataset using five-fold cross-validation on the ML evaluation metrics (Figure 4 in the Appendix). Second, we optimized the hyper-parameters for each combination of DGM, dataset, and loss function (Figure 2).

## 3.4 ITERATIVE OBJECTIVE REFINEMENT BAYESIAN OPTIMIZATION

Previous research on DNN often relied on tuning hyper-parameters based on a single metric or aggregating multiple metrics with varying units in SBO. For example, the objective function guiding the BO process could be the Dice score for medical segmentation (Vu et al., 2021), mean macro-

---

[1]https://hyperopt.github.io/hyperopt/

accuracy for visual question answering (Vu et al., 2020), or metrics like F-score (classification) and $R$-squared (regression) evaluated with Catboost (Dorogush et al., 2018) on synthetic tabular data (Kotelnikov et al., 2023). A significant challenge in SBO arises from managing diverse metrics, such as those used in statistical evaluations and ML performance, that differ in units, complicating direct aggregation. This limitation can hinder the ability to fully capture trade-offs between different objectives. To overcome the issues associated with aggregating metrics with varying units in multi-objective SBO, we propose a ranking-based approach, named IORBO, to enhance BO performance.

To illustrate, consider optimizing a DGM. We define $\boldsymbol{y}_u$ as the vector comprising all evaluated metrics where $u \in \{1, \ldots, U\}$, with $U$ representing the number of samples used in the optimization. In the SBO, the objective function of sample $u$ is defined as $r_u = f(\boldsymbol{y}_u)$ where $f$ is an aggregation function. As outlined in Algorithm 1, the SBO holds $r_u$ constant throughout the optimization.

In contrast, IORBO defines the objective function as $r_u^{(p)}$, where $u \leq p$ and $p \in \{1, \ldots, U\}$ (see Algorithm 2). Here, $u$ represents the iteration where the objective is first generated, while $p$ denotes when it is updated, introducing iterative refinement into the process. In the IORBO, the objective function of sample $u$ is defined as $r_u^{(p)} = g(\boldsymbol{y}_u | \boldsymbol{y}_1, \boldsymbol{y}_2, \ldots, \boldsymbol{y}_p)$ where $g$ is a rank-based function. For example, at the second iteration, $\boldsymbol{y}_2$ is evaluated, then both $r_1^{(2)}$ and $r_2^{(2)}$ are computed. In the third iteration, $\boldsymbol{y}_3$ is added, allowing for the computation of $r_1^{(3)}, r_2^{(3)}$, and $r_3^{(3)}$, and so on. The objective functions are recalculated as the mean ranks of all generated samples, yielding $r_1^{(u)}, r_2^{(u)}, \ldots, r_u^{(u)}$ based on $\boldsymbol{y}_1, \boldsymbol{y}_2, \ldots, \boldsymbol{y}_u$. To compute the mean ranks, all data points that are generated by the IORBO for each evaluated metric are first ranked and then the average rank across metrics is calculated.

The objective function for the first set of hyper-parameters, $\Theta_1$, is iteratively updated: $r_1^{(1)} \to r_1^{(2)} \to \cdots \to r_1^{(U)}$. For the $\Theta_2$, we updated: $r_2^{(2)} \to r_2^{(3)} \to \cdots \to r_2^{(U)}$, and so on. The surrogate model is simultaneously refitted with the revised samples, $(\Theta_1, r_1^{(u)}), (\Theta_2, r_2^{(u)}), \ldots, (\Theta_u, r_u^{(u)})$. IORBO incurs a slight additional cost for refitting the surrogate model with revised samples during the iterative refinement. However, this overhead is negligible compared to the overall computational cost. Apart from this refinement step, the process is essentially the same as SBO. For a numerical illustration, see Section E in the Appendix.

| **Algorithm 1** Standard Bayesian Optimization (SBO) | **Algorithm 2** Iterative Objective Refinement Bayesian Optimization (IORBO) |
|---|---|
| Initialize surrogate model | Initialize surrogate model |
| Initialize generative model $\mathcal{G}_i$ | Initialize generative model $\mathcal{G}_i$ |
| Suggest initial hyper-parameters $\Theta_1$ | Suggest initial hyper-parameters $\Theta_1$ |
| Build and train $\mathcal{G}_i$ | Build and train $\mathcal{G}_i$ |
| Perform evaluation to obtain $\boldsymbol{y}_1$ | Perform evaluation to obtain $\boldsymbol{y}_1$ |
| Compute $r_1 = f(\boldsymbol{y}_1)$ | Compute $r_1^{(1)} = g(\boldsymbol{y}_1 \| \boldsymbol{y}_1)$ |
| Fit surrogate model with $(\Theta_1, r_1)$ | Fit surrogate model with $(\Theta_1, r_1^{(1)})$ |
| **for** $u \leftarrow 2$ to $U$ **do** | **for** $u \leftarrow 2$ to $U$ **do** |
|     Suggest $\Theta_u$ |     Suggest $\Theta_u$ |
|     Build and train $\mathcal{G}_i$ |     Build and train $\mathcal{G}_i$ |
|     Perform evaluation to obtain $\boldsymbol{y}_u$ and $r_u$ |     Perform evaluation to obtain $\boldsymbol{y}_u$ |
|     Update surrogate model with $(\Theta_u, r_u)$ |     Update ranks $\{r_1^{(u)}, r_2^{(u)}, \cdots, r_u^{(u)}\}$ based on $\{\boldsymbol{y}_1, \boldsymbol{y}_2, \cdots, \boldsymbol{y}_u\}$ |
| |     Fit surrogate model with revised samples $(\Theta_1, r_1^{(u)}), (\Theta_2, r_2^{(u)}), \ldots, (\Theta_u, r_u^{(u)})$ |
| **end for** | **end for** |
| **return** Optimal hyper-parameters $\Theta^*$ | **return** Optimal hyper-parameters $\Theta^*$ |

## 3.5 STATISTICAL TESTS

To compare loss functions across DGMs and datasets, we used the Friedman test (Friedman, 1937; 1940) to rank the loss functions independently. For non-parametric analysis of repeated-measures

data, the Friedman test offers an alternative to the widely used repeated-measures ANOVA (Fisher, 1919). We used the Friedman test with equivalence on two ML efficacy problems for test set predictions and statistical similarity between training and synthetic data (detailed in Section 3.6). Following Demšar (2006), we further explored significant differences between methods using the Nemenyi post-hoc test (Nemenyi, 1963). Table 1 shows the $p$-values divided into three positive and three negative differences.

## 3.6 BENCHMARKING FRAMEWORK

Figure 3 provides an overview of the proposed benchmarking framework, which consists of the following core components:

*Generative models.* DGMs are used to generate synthetic data. We evaluated six models. Three models that leverage conditional GANs for data synthesis: `CTGAN` (Xu et al., 2019), `CTAB-GAN` (Zhao et al., 2021), and

Table 1: Ranges of $p$-values and specification obtained from statistical tests.

| Notation | Rank | Range of $p$-value | Specification |
|---|---|---|---|
| $++$ | Better | $p \leq 0.01$ | Highly significantly better |
| $+$ | Better | $0.01 < p \leq 0.05$ | Significantly better |
| $0$ | Better | $p > 0.05$ | Not significantly better |
| $0$ | Worse | $p > 0.05$ | Not significantly worse |
| $-$ | Worse | $0.01 < p \leq 0.05$ | Significantly worse |
| $--$ | Worse | $p \leq 0.01$ | Highly significantly worse |

`DP-CGANS` (Sun et al., 2023). A model that combines Gaussian Copula with the `CTGAN` architecture: `CopulaGAN`. A model that utilizes VAEs (Kingma and Welling, 2013) for data generation: `TVAE` (Xu et al., 2019). Finally, a model that employs DDPM: `TabDDPM` (Kotelnikov et al., 2023). To explore the impact of the conditional element, we additionally evaluated versions of `CTGAN`, `CopulaGAN`, and `DP-CGANS` with conditioning disabled. We also used two backbones for `TabDDPM`: a simple multilayer perceptron (MLP) and a ResNet.

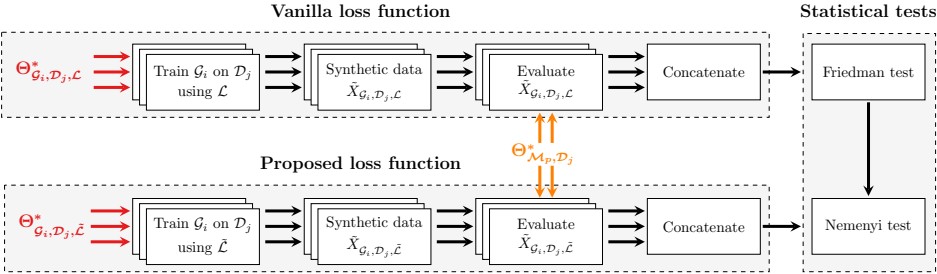

Figure 3: Proposed benchmarking framework. The $\mathcal{G}_i$, $\mathcal{D}_j$, $\mathcal{L}_k$ and $\mathcal{M}_p$ denote a DGM, a dataset, a loss function, and an ML method, respectively. $\Theta^*$ denotes the optimal set of hyper-parameters. See Figure 2 and Figure 4 in the Appendix to see how we determined $\Theta^*_{\mathcal{G}_i,\mathcal{D}_j,\mathcal{L}_k}$ and $\Theta^*_{\mathcal{M}_p,\mathcal{D}_j}$.

*Custom loss function.* During training, each evaluated DGM utilized either the custom loss function defined in Equation 8 (for GAN models), the one presented in Equation 9 (for `TVAE` model) or the one in Equation 10 (for `TabDDPM` model). We subsequently fixed $\alpha$, $\beta$, and $\zeta$ to specific values of 0 or positive value, resulting in two different experiments: vanilla loss function ($\mathcal{L}$ with $\alpha = \beta = \zeta = 0$) and the proposed loss function, $\widetilde{\mathcal{L}}$, with at least one non-zero hyper-parameters.

*Statistical tests.* We used the Friedman test on all evaluated metrics, followed by the Nemenyi post-hoc test detailed in Section 3.5 for comparative analyses. These analyses can be divided into three categories: (1) General-purpose loss function assesses which loss function—between the vanilla (original loss function used in the evaluated DGM) and the proposed—performs better for general applications; (2) Dataset-specific loss determines which loss function is more effective for each evaluated dataset; and (3) Method-specific loss identifies the superior loss function for each evaluated DGM architecture. For each category, we based the evaluations on either statistical similarity, ML TSTR performance, ML augmentation performance, or a combination of evaluated metrics.

**Loss function.** To analyze the performance of the proposed loss function against the vanilla version, we applied the benchmarking framework using different independent evaluations: (1) statistical

analysis on its own, (2) ML TSTR performance on its own, (3) ML augmentation on its own, and (4) a comprehensive evaluation that combines all evaluated metrics.

**Bayesian Optimization Method.** To compare the performance of the IORBO with the SBO using mean and median aggregation methods, we fine-tuned each DGM on each dataset across different loss functions, employing three evaluated BO approaches. Statistical tests were then conducted to evaluate the three BO methods.

## 4 EXPERIMENTS

*Datasets.* To evaluate the capability of the proposed method, we conducted experiments on twenty publicly available datasets encompassing a variety of ML tasks, data sizes, and diversities in terms of categorical and continuous variables (detailed in Table 6 in the Appendix).

*Implementation Details and Training.* We implemented all DGMs (`CTGAN`, `CTAB-GAN`, `DP-CGANS`, `CopulaGAN`, `TVAE`, and `TabDDPM`) and the proposed losses using PyTorch 1.13. To ensure replicability, we maintained the DGMs' original framework structures and adopted the model parameters specified in their publications. We disabled conditional elements within evaluated DGMs by reimplementing their data samplers. This modification removed the conditional vector from the training process, effectively transforming them into unconditional DGMs. For all DGMs, we employed the Adam optimizer (Kingma and Ba, 2015). We used the proposed IORBO approach introduced in Section 3.4 to fine-tune the hyper-parameters in two tuning processes (Section 3.3). See Section D in the Appendix for more details on the implementation and training. The detailed search spaces are provided in Section H in the Appendix.

## 5 RESULTS AND DISCUSSION

**Loss function.** To analyze the performance of the proposed loss function against the vanilla version, we employed the proposed benchmarking framework (Section 3.6) across four key tasks: statistical evaluation (Stat.), TSTR evaluation, augmentation evaluation (Aug.), and a comprehensive evaluation (Comp.) combining all three. The statistical tests evaluated the performance of the proposed loss function compared to the vanilla loss. In addition, we define the *win rate* as as the proportion of evaluated metrics where the proposed loss function exceeds the vanilla loss function, relative to the total number of metrics assessed. A win rate of 1 indicates that the proposed loss function performed better than the vanilla version across all evaluated metrics, while a value of 0 signifies that it performed worse in every metric. A win rate greater than 0.5 indicates that the proposed loss function was "better" more often than it was "worse." We also report standard errors for each metric, estimated from 1 000 bootstrap rounds.

Table 2: Results of the Nemenyi post-hoc test and win rate (with standard error in parentheses) comparing the proposed against the vanilla loss function on all DGMs and datasets. Loss functions were evaluated for statistical similarity (Stat.), TSTR, augmentation (Aug.), and a comprehensive evaluation (Comp.) combining all metrics. For details on $p$-value ranges, refer to Table 1.

| | Statistical Tests | | | | Win Rate | | | |
|---|---|---|---|---|---|---|---|---|
| Comparison | Stat. | TSTR | Aug. | Comp. | Stat. | TSTR | Aug. | Comp. |
| Proposed vs. Vanilla | 0 | ++ | ++ | ++ | 0.484 (0.012) | **0.611 (0.007)** | **0.551 (0.007)** | **0.567 (0.004)** |

*General-purpose loss function.* Table 2 presents the results of a comprehensive analysis comparing the performance of the proposed loss function against the vanilla loss function across all DGMs and datasets. The table highlights the influence of loss function selection for general purposes.

First, two loss functions performed statistically similarly (zero (0) in the "Stat." column in Table 2). Second, in the ML TSTR evaluation, the proposed loss function significantly outperformed the vanilla version, with a win rate of 0.611 and a standard error of 0.007, suggesting that the proposed loss function better captures the complexities of real-world tabular data during synthetic data generation. Third, the augmentation evaluation consistently favored the proposed loss function (win rate 0.551), demonstrating its ability to enhance the performance of predictive models trained on a mix of real and synthetic data. Finally, the comprehensive evaluation (win rate 0.567), which com-

bined all prior evaluations, continues this trend, indicating the proposed loss function's potential to improve model generalizability. A possible reason for this superiority is that the proposed loss function provides a regularizing effect, which likely reduces overfitting on unseen data and positions it as a strong candidate for general-purpose use in generative modeling tasks.

Table 3: Results of the Nemenyi post-hoc test and win rate (with standard error in parentheses) comparing the proposed against the vanilla loss function across various DGMs on all datasets. Evaluations include TSTR, augmentation (Aug.), statistical similarity (Stat.), and a comprehensive measure (Comp.) combining all evaluated metrics. Models denoted with an asterisk (*) have disabled conditioning. For details on $p$-value ranges, refer to Table 1.

| | Statistical Tests | | | | Win Rate | | | |
|---|---|---|---|---|---|---|---|---|
| Method | Stat. | TSTR | Aug. | Comp. | Stat. | TSTR | Aug. | Comp. |
| CTGAN | 0 | ++ | ++ | ++ | 0.478 (0.034) | **0.639 (0.020)** | **0.583 (0.021)** | **0.593 (0.014)** |
| CTGAN⋆ | 0 | ++ | ++ | ++ | 0.459 (0.036) | **0.726 (0.018)** | **0.611 (0.020)** | **0.640 (0.013)** |
| TVAE | 0 | 0 | ++ | ++ | **0.519 (0.034)** | **0.501 (0.021)** | **0.593 (0.020)** | **0.543 (0.013)** |
| CopulaGAN | 0 | ++ | + | ++ | 0.491 (0.033) | **0.633 (0.020)** | **0.547 (0.022)** | **0.577 (0.013)** |
| CopulaGAN⋆ | 0 | ++ | + | ++ | 0.447 (0.034) | **0.684 (0.019)** | **0.554 (0.020)** | **0.595 (0.013)** |
| DP-CGANS | 0 | ++ | ++ | ++ | 0.500 (0.051) | **0.669 (0.028)** | **0.683 (0.028)** | **0.651 (0.018)** |
| DP-CGANS⋆ | 0 | ++ | 0 | ++ | **0.587 (0.054)** | **0.798 (0.023)** | **0.538 (0.030)** | **0.656 (0.019)** |
| CTAB-GAN | −− | −− | 0 | −− | 0.391 (0.033) | 0.418 (0.020) | 0.497 (0.020) | 0.448 (0.014) |
| TABDDPM-MLP | 0 | ++ | 0 | ++ | **0.516 (0.035)** | **0.617 (0.020)** | 0.482 (0.021) | **0.545 (0.014)** |
| TABDDPM-ResNet | 0 | + | 0 | 0 | **0.512 (0.033)** | **0.547 (0.021)** | 0.487 (0.021) | **0.517 (0.013)** |

*Method-specific loss function.* Table 3 compares the performance of the proposed loss function against the vanilla loss functions across all datasets and different DGM selections. Models denoted with an asterisk (*) have disabled conditioning. For most models, the proposed loss function demonstrates significant improvements in ML TSTR performance and augmentation effectiveness. For instance, CTGAN, CTGAN⋆, CopulaGAN, and DP-CGANS consistently show highly significant gains (++) in TSTR, augmentation, and comprehensive evaluation. For example, DP-CGANS⋆ achieved the highest win rate across the TSTR metric, 0.798, indicating that the proposed loss function significantly enhanced its ability to generate synthetic data that boosts downstream ML performance.

Interestingly, the statistical similarity (Stat.) evaluation reveals no significant differences between the proposed and vanilla loss functions for most models, suggesting that both loss functions perform similarly in terms of generating synthetic data that statistically match the real data distributions. However, the CTAB-GAN model stands out as an exception, showing a statistically significant decrease (−−) in performance across most evaluations when using the proposed loss function. This result suggests that the CTAB-GAN may require a more specialized loss function or optimization strategy to fully benefit from the proposed approach.

The comprehensive evaluation (Comp.), which combines all three metrics, underscores the effectiveness of the proposed loss function on eight out of ten evaluated DGMs. Models including CTGAN, CopulaGAN, DP-CGANS, and their non-conditioned variants, consistently outperform the vanilla loss function with win rates exceeding 0.5. These results imply that the proposed loss function offers a well-rounded improvement across various aspects of synthetic data generation, specifically in terms of enhancing ML utility and model augmentation performance.

*Dataset-specific loss function.* Table 4 compares the proposed loss function to the vanilla loss function across various datasets on all DGMs. The results demonstrate the effectiveness of the proposed loss function across a diverse set of datasets. The statistical tests reveal that the proposed loss function achieves statistically significant improvements in TSTR performance for 14 out of 20 datasets, as indicated by the total count of (+) and (++). In addition, the proposed loss function exhibits a consistent advantage in augmentation (Aug.). Specifically, datasets such as Insurance and MNIST12 show marked improvements in win rates (0.7). Conversely, the proposed loss function shows variable performance in statistical similarity (Stat.) across different datasets. While it significantly improves TSTR and augmentation tasks for many datasets, its impact on statistical similarity is less consistent, with some datasets like Cardio, Higgs-Small, and Miniboone exhibiting inferior results compared to the vanilla loss function.

From Table 4 we see that the proposed loss function demonstrates significant improvement over the vanilla loss function in 15 out of 20 datasets, as indicated by the comprehensive evaluation (Comp.) in the statistical tests column. Among the remaining datasets, four show no significant difference (0) and only one shows a statistically significant disadvantage (−).

Table 4: Results of the Nemenyi post-hoc test and win rate (with standard error in parentheses) comparing the proposed against the vanilla loss function across various datasets on all evaluated DGMs. Evaluations include TSTR, augmentation (Aug.), statistical similarity (Stat.), and a comprehensive measure (Comp.) combining all three. For details on $p$-value ranges, refer to Table 1.

| Dataset | Statistical Tests | | | | Win Rate | | | |
|---|---|---|---|---|---|---|---|---|
| | Stat. | TSTR | Aug. | Comp. | Stat. | TSTR | Aug. | Comp. |
| Abalone | 0 | ++ | −− | 0 | **0.594 (0.059)** | **0.633 (0.043)** | 0.375 (0.045) | **0.523 (0.027)** |
| Adult | 0 | ++ | 0 | ++ | **0.538 (0.052)** | **0.622 (0.025)** | **0.553 (0.025)** | **0.582 (0.017)** |
| Buddy | 0 | ++ | 0 | ++ | 0.500 (0.051) | **0.607 (0.022)** | **0.552 (0.022)** | **0.570 (0.014)** |
| California | 0 | 0 | 0 | + | 0.500 (0.045) | **0.583 (0.044)** | **0.583 (0.043)** | **0.566 (0.027)** |
| Cardio | − | ++ | ++ | ++ | 0.387 (0.054) | **0.577 (0.027)** | **0.590 (0.027)** | **0.560 (0.019)** |
| Churn2 | 0 | ++ | 0 | + | 0.500 (0.059) | **0.637 (0.025)** | 0.460 (0.026) | **0.543 (0.018)** |
| Credit | 0 | 0 | 0 | + | 0.500 (0.057) | **0.544 (0.027)** | **0.554 (0.030)** | **0.543 (0.018)** |
| Diabetes | 0 | ++ | 0 | ++ | 0.413 (0.031) | **0.620 (0.027)** | **0.533 (0.027)** | **0.557 (0.018)** |
| Diabetes-ML | 0 | ++ | 0 | ++ | 0.469 (0.057) | **0.719 (0.029)** | 0.479 (0.033) | **0.584 (0.020)** |
| Diabetes Bal. | 0 | ++ | 0 | ++ | 0.438 (0.032) | **0.717 (0.022)** | **0.538 (0.025)** | **0.605 (0.016)** |
| Gesture | 0 | 0 | 0 | 0 | **0.609 (0.056)** | **0.562 (0.032)** | 0.450 (0.029) | **0.518 (0.021)** |
| Higgs-Small | − | + | ++ | ++ | 0.359 (0.055) | **0.575 (0.031)** | **0.635 (0.032)** | **0.576 (0.021)** |
| House | 0 | ++ | ++ | ++ | 0.438 (0.049) | **0.667 (0.039)** | **0.667 (0.038)** | **0.618 (0.025)** |
| House-16h | 0 | 0 | 0 | 0 | 0.500 (0.044) | 0.442 (0.044) | 0.500 (0.046) | 0.477 (0.027) |
| Insurance | 0 | ++ | ++ | ++ | 0.494 (0.052) | **0.693 (0.037)** | **0.700 (0.037)** | **0.654 (0.025)** |
| King | 0 | ++ | 0 | ++ | **0.519 (0.055)** | **0.673 (0.039)** | **0.567 (0.039)** | **0.599 (0.026)** |
| Miniboone | − | −− | 0 | − | 0.359 (0.054) | 0.402 (0.031) | **0.512 (0.026)** | 0.446 (0.020) |
| MNIST12 | 0 | ++ | ++ | ++ | 0.484 (0.040) | **0.756 (0.023)** | **0.700 (0.024)** | **0.699 (0.016)** |
| News | + | ++ | 0 | ++ | **0.612 (0.052)** | **0.607 (0.040)** | **0.553 (0.042)** | **0.587 (0.024)** |
| Wilt | 0 | 0 | 0 | 0 | 0.469 (0.056) | **0.538 (0.025)** | **0.552 (0.026)** | **0.536 (0.016)** |

Table 5: Results of the Nemenyi post-hoc test and win rate (with standard error in parentheses) comparing the row to column method. For details on $p$-value ranges, refer to Table 1.

| BO method | Statistical Tests | | | Win Rate | | |
|---|---|---|---|---|---|---|
| | IORBO | SBO-Mean | SBO-Median | IORBO | SBO-Mean | SBO-Median |
| IORBO | | ++ | ++ | | **0.591 (0.004)** | **0.561 (0.004)** |
| SBO-Mean | −− | | −− | 0.409 (0.004) | | 0.461 (0.004) |
| SBO-Median | −− | ++ | | 0.439 (0.004) | **0.539 (0.004)** | |

**Bayesian optimization method.** The performance of the IORBO was compared to the SBO using two aggregation methods (mean and median aggregation). We fine-tuned each DGM on each dataset across two loss functions, and employed three evaluated BO approaches. Statistical tests were then conducted to evaluate the three BO methods. Table 5 shows the results of the Nemenyi post-hoc test and win rate (with standard error in parentheses) comparing methods in the rows to those in the columns. The Nemenyi post-hoc test indicates that the IORBO is significantly better than the SBO-Mean and SBO-Median with win rates of 0.591 and 0.561, respectively. The results demonstrate that the IORBO is robust in handling metrics with different units and its potential as a reliable, broadly applicable BO method.

Due to space limitations, further details on the ablation studies are presented in Section F. These studies emphasize the critical role of both the proposed loss function and IORBO in enhancing model performance, with the combination of the two consistently yielding the best results across different configurations.

## 6 CONCLUSION

We have introduced a novel correlation- and distribution-aware loss function designed as a regularizer for DGMs in tabular data synthesis, which outperforms the vanilla loss function across most DGMs. The results suggest that the proposed loss function effectively captures the complexities of arbitrary DGMs. Future research could focus on addressing potential numerical instability when incorporating higher-order moments by using an exponential moving average of the moments over iterations, ensuring the moments match on average rather than for a single mini-batch, as well as on developing a tailored loss function for the `CTAB-GAN` family to match the strong performance seen with other DGMs. Additionally, we introduced a novel IORBO approach that leverages rank-based aggregation to ensure more meaningful comparisons between multiple objectives with varying units, providing a more robust optimization process. Finally, we developed a comprehensive benchmarking system evaluating statistical similarity, ML TSTR performance, and ML augmentation performance, with robust statistical tests, offering a valuable tool for future research.

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

# APPENDIX

## A EVALUATION

*Statistical similarity evaluation.* We employed four key metrics to quantify how closely the real and synthetic data distributions resemble each other. (1) The KL divergence (Hershey and Olsen, 2007): This method quantifies the information loss incurred when approximating a true probability distribution with another one. (2) The Pearson's CS test (Pearson, 1992): This test focuses on categorical variables and assesses whether the distribution of categories in the synthetic data matches the distribution in the real data. (3) The KS test (Massey Jr, 1951): This test is designed for continuous variables and measures the distance between the cumulative distribution functions (CDFs) of the real and synthetic data. (4) The DWP: We leveraged the DWP (Armanious et al., 2020) to quantitatively assess the quality of the generated data. This metric evaluates how well the model captures the distribution of each individual class or variable. To calculate the DWP metric, we compute the average distance between scatter points and a perfect diagonal line ($y = x$). Each scatter point represents either a class within a categorical variable or the mean value of a continuous variable.

To assess how effectively the synthetic data captures the inherent relationships between variables observed in the real data, we compare correlation coefficients between variable pairs. For continuous variables, we employ the widely-used Pearson correlation coefficient, calculated from both the real and synthetic data matrices. In the case of categorical variables, we leverage Cramer's V coefficient to quantify the association strength between each pair in both datasets (Frey, 2018).

To assess the ML performance in both the TSTR and augmentation tasks, we split the experimental datasets into 80% training and 20% testing sets. First, we trained the DGMs on the real training data to produce synthetic data. The real testing set served a critical role in assessing the generalizability of trained ML models on unseen data. Subsequently, for TSTR, we trained various ML methods including logistic regression (LG), support vector machine (SVM), random forest (RF), bagging (bootstrap aggregating), and XGBoost independently on both the real and synthetic training sets. In the augmentation task, we trained the same ML models independently on both the real training set and a combined set consisting of real training and synthetic data.

## B DATASETS

Two datasets come from the UCI Machine Learning Repository (Dua and Graff, 2017) (`Adult` and `News`) and feature tabular structures with separate columns for attributes and labels. Thirteen additional datasets were preprocessed and shared by Kotelnikov et al. (2023) including `Abalone`, `Buddy`, `California`, `Cardio`, `Churn2`, `Diabetes-ML`, `Gesture`, `Higgs-Small`, `House-16h`, `Insurance`, `King`, `Miniboone`, and `Wilt`. We sourced the remaining datasets from Kaggle[2] (`Credit`, `Diabetes`, `Balanced Diabetes`, and `House`). To investigate the method's behavior on high-dimensional binary data as in (Xu et al., 2019), we transformed the Modified National Institute of Standards and Technology database (MNIST) dataset (LeCun and Cortes, 2010). Specifically, we binarized the original $28 \times 28$ images, converted each sample into a 784-dimensional vector, and added a label column. The images were then resized to $12 \times 12$, reducing them to 144-dimensional vectors. We refer to this dataset as `MNIST12`.

Table 6 provides a comprehensive overview of the datasets evaluated in our study. It includes a diverse set of datasets, encompassing various data types and tasks to thoroughly test the proposed methods. The datasets range from small, specialized datasets like `Diabetes-ML` with 768 rows and 8 continuous variables, to large, extensive datasets such as `Credit` with 277 640 rows and 29 continuous variables. Tasks represented include regression, binary classification, and multiclass classification, showcasing the breadth of application scenarios covered. For instance, `Abalone` and `California` are used for regression tasks, while `Adult`, `Cardio`, and `Churn2` are employed for binary classification tasks. Multiclass classification tasks are represented by datasets such as `Buddy` and `MNIST12`.

Additionally, the datasets exhibit a range of characteristics in terms of the number of continuous and discrete variables. For example, `Gesture` has a high number of continuous variables (32) with no

---

[2]https://www.kaggle.com/datasets

Table 6: Description of experimented datasets. "Cat." and "Cont." stand for categorical and continuous variables, respectively. "Classif." denotes classification, while "Reg." is regression.

| Dataset | #Rows | #Cont. | #Dis. | Task |
|---|---|---|---|---|
| Abalone | 4 177 | 7 | 1 | Regression |
| Adult | 48 813 | 6 | 8 | Binclass |
| Buddy | 18 834 | 4 | 5 | Multiclass |
| California | 20 640 | 8 | 0 | Regression |
| Cardio | 70 000 | 5 | 6 | Binclass |
| Churn2 | 10 000 | 7 | 4 | Binclass |
| Credit | 277 640 | 29 | 0 | Binclass |
| Diabetes | 234 245 | 0 | 21 | Binclass |
| Diabetes-ML | 768 | 8 | 0 | Binclass |
| Diabetes Bal. | 69 515 | 0 | 21 | Binclass |
| Gesture | 9 873 | 32 | 0 | Multiclass |
| Higgs-Small | 98 049 | 28 | 0 | Binclass |
| House | 21 613 | 10 | 8 | Regression |
| House-16h | 22 784 | 16 | 0 | Regression |
| Insurance | 1 338 | 3 | 3 | Regression |
| King | 21 613 | 17 | 3 | Regression |
| Miniboone | 130 064 | 50 | 0 | Binclass |
| MNIST12 | 70 000 | 0 | 144 | Multiclass |
| News | 39 644 | 45 | 14 | Regression |
| Wilt | 4 839 | 5 | 0 | Binclass |

discrete variables, whereas `Diabetes` features a substantial number of discrete variables (21) with no continuous variables. The varied nature of these datasets allows for a robust evaluation of the proposed methods across different types of data and tasks, providing insights into their generalizability and effectiveness. The inclusion of datasets with different characteristics, such as `Higgs-Small` with 28 continuous variables and `MNIST12` with 144 discrete variables, ensures a comprehensive assessment of performance and applicability.

## C    STATISTICAL TESTS

In Table 1, the specifications are based on the commonly accepted interpretation of $p$-values in hypothesis testing. A $p$-value less than or equal to 0.01 ($p \leq 0.01$) indicates that the result is highly significant, meaning that the null hypothesis can be rejected with high confidence. A $p$-value between 0.01 and 0.05 ($0.01 < p \leq 0.05$) indicates significant results, where there is still a reasonable level of evidence against the null hypothesis, though not as strong as for the highly significant results. For $p$-values greater than 0.05, we consider the result not to be significant, indicating insufficient evidence to reject the null hypothesis.

Regarding the two-sided test, the Nemenyi post-hoc test used in our analysis is based on the Friedman test, which is a non-parametric test for repeated measures. The Nemenyi test performs pairwise comparisons between the groups following the Friedman test and is a two-sided test. This means that the test evaluates whether the differences between the groups are statistically significant in both directions, *i.e.*, it considers whether one group is significantly better or worse than another group.

## D    IMPLEMENTATION DETAILS AND TRAINING

The experiments ran on a high-performance computing cluster equipped with NVIDIA A100 Tensor Core graphical processing units (GPUs) (40GB RAM each) and Intel(R) Xeon(R) Gold 6338 CPUs (256GB DDR4 RAM). Training time per model varied significantly by dataset and DGM, ranging from one hour to two weeks.

To accelerate the ML performance evaluation, we used the `cuML` library (Raschka et al., 2020). This library provides a Python API largely compatible with `scikit-learn` (Pedregosa et al., 2011) and allows seamless execution of traditional tabular ML tasks on GPUs. We used `scikit-learn` for classification and regression metrics, `scipy` for statistical evaluation metrics, and `scikit-posthocs` for the statistical tests, ensuring consistency throughout the evaluation process.

## E    Illustrative Example: SBO and IORBO in Practice Comparison

Table 7 and Table 8 illustrate three iterations of SBO and IORBO, respectively. Evaluated metrics are $\boldsymbol{y} = \{a, b, c, d\}$. In Table 7, we see that $r_1 = r_2 = r_3$. However, Table 8 shows that in the IORBO objective functions differ after three iterations: $r_1^{(3)} \neq r_2^{(3)} \neq r_3^{(3)}$ and $r_1^{(1)} \neq r_1^{(2)} \neq r_1^{(3)}$.

Table 7: Example of SBO where the objective function is computed as the mean of all evaluated metrics.

| | Iteration 1 | Iteration 2 | | Iteration 3 | | |
|---|---|---|---|---|---|---|
| Metric / Sample | 1 | 1 | 2 | 1 | 2 | 3 |
| $a$ | 1 | 1 | 0.5 | 1 | 0.5 | 2.4 |
| $b$ | 1 | 1 | 2.5 | 1 | 2.5 | 0.2 |
| $c$ | 1 | 1 | 0.5 | 1 | 0.5 | 0.8 |
| $d$ | 1 | 1 | 0.5 | 1 | 0.5 | 0.6 |
| Objective function | $r_1 = 1$ | $r_1 = 1$ | $r_2 = 1$ | $r_1 = 1$ | $r_2 = 1$ | $r_3 = 1$ |

Table 8: Example of IORBO.

| | Iteration 1 | Iteration 2 | | Iteration 3 | | |
|---|---|---|---|---|---|---|
| Metric / Sample | 1 | 1 | 2 | 1 | 2 | 3 |
| $a$ | 1 | 1 | 0.5 | 1 | 0.5 | 2.4 |
| $b$ | 1 | 1 | 2.5 | 1 | 2.5 | 0.2 |
| $c$ | 1 | 1 | 0.5 | 1 | 0.5 | 0.8 |
| $d$ | 1 | 1 | 0.5 | 1 | 0.5 | 0.6 |
| Metric ranking / Sample | 1 | 1 | 2 | 1 | 2 | 3 |
| $a$ | 1 | 2 | 1 | 2 | 1 | 3 |
| $b$ | 1 | 1 | 2 | 2 | 3 | 1 |
| $c$ | 1 | 2 | 1 | 3 | 1 | 2 |
| $d$ | 1 | 2 | 1 | 3 | 1 | 2 |
| Objective function | $r_1^{(1)} = 1$ | $r_1^{(2)} = 1.75$ | $r_2^{(2)} = 1.25$ | $r_1^{(3)} = 2.5$ | $r_2^{(3)} = 1.5$ | $r_3^{(3)} = 2$ |

## F    Ablation Studies

The ablation studies presented in Table 9 and Table 10 investigate the impact of combining vanilla and proposed loss functions with two optimization strategies: SBO and IORBO, across all evaluated DGMs and datasets. Table 9 focuses on evaluating SBO-Mean and IORBO with both loss functions, while Table 10 examines similar combinations for SBO-Median and IORBO. These analyses aim to isolate the contributions of each component—optimization method and loss function—to overall model performance.

The results from both tables, supported by the Nemenyi post-hoc tests, consistently show that the IORBO + Proposed configuration outperforms the others across both mean and median evaluations. Specifically, in Table 9, IORBO + Proposed achieves win rates of 0.644 (vs. SBO-Mean + Vanilla), 0.582 (vs. IORBO + Vanilla), and 0.601 (vs. SBO-Mean + Proposed), significantly outperforming all other configurations. Likewise, in Table 10, IORBO + Proposed also leads with win rates of 0.600, 0.582, and 0.577, further demonstrating the synergy between the proposed loss function and IORBO optimization. These results suggest that the proposed loss function significantly enhances the model's capacity to adapt to the data distribution when combined with IORBO.

In Table 9, comparing SBO-Mean + Vanilla to SBO-Mean + Proposed shows the contribution of the proposed loss function to performance improvement. The win rate for SBO-Mean + Proposed (0.581) against SBO-Mean + Vanilla underlines the importance of the loss function in enhancing quality of synthetic data. Similarly, comparing SBO-Mean + Vanilla to IORBO + Vanilla reveals that the incorporation of IORBO optimization improves performance significantly. This indicates that the proposed IORBO contributes effectively to better results.

The same trend is observed in Table 10, where comparisons between SBO-Median + Vanilla and SBO-Median + Proposed, as well as between SBO-Median + Vanilla and IORBO + Vanilla, demonstrate similar performance gains.

In conclusion, these findings underline the critical roles of both the proposed loss function and the IORBO optimization method in enhancing model performance. The combination of these elements in IORBO + Proposed consistently leads to the best outcomes, validating the effectiveness of integrating both components for superior performance in optimization tasks.

Table 9: Results of the Nemenyi post-hoc test and win rate (with standard error in parentheses) comparing row and column methods. The table presents performance across different configurations, including the baseline with SBO and mean aggregation with the vanilla loss function, and comparisons with the proposed loss function and IORBO optimization method. For details on $p$-value ranges, refer to Table 1. "Van." and "Prop." denote the vanilla and proposed loss functions, respectively.

| Method | Statistical Tests | | | | Win Rate | | | |
|---|---|---|---|---|---|---|---|---|
| | SBO-Mean + Van. | IORBO + Van. | SBO-Mean + Prop. | IORBO + Prop. | SBO-Mean + Van. | IORBO + Van. | SBO-Mean + Prop. | IORBO + Prop. |
| SBO-Mean + Van. | | −− | −− | −− | | 0.420 (0.004) | 0.419 (0.004) | 0.356 (0.003) |
| IORBO + Van. | ++ | | ++ | −− | 0.580 (0.004) | | 0.525 (0.004) | 0.418 (0.004) |
| SBO-Mean + Prop. | ++ | −− | | −− | 0.581 (0.004) | 0.475 (0.004) | | 0.399 (0.004) |
| IORBO + Prop. | ++ | ++ | ++ | | 0.644 (0.003) | 0.582 (0.004) | 0.601 (0.004) | |

Table 10: Results of the Nemenyi post-hoc test and win rate (with standard error in parentheses) comparing row and column methods. The table presents performance across different configurations, including the baseline with SBO and median aggregation with the vanilla loss function, and comparisons with the proposed loss function and IORBO optimization method. For details on $p$-value ranges, refer to Table 1. "Van." and "Prop." denote the vanilla and proposed loss functions, respectively.

| Method | Statistical Tests | | | | Win Rate | | | |
|---|---|---|---|---|---|---|---|---|
| | SBO-Med. + Van. | IORBO + Van. | SBO-Med. + Prop. | IORBO + Prop. | SBO-Med. + Van. | IORBO + Van. | SBO-Med. + Prop. | IORBO + Prop. |
| SBO-Med. + Van. | | −− | −− | −− | | 0.454 (0.004) | 0.458 (0.004) | 0.400 (0.003) |
| IORBO + Van. | ++ | | 0 | −− | 0.546 (0.004) | | 0.503 (0.004) | 0.418 (0.004) |
| SBO-Med. + Prop. | ++ | 0 | | −− | 0.542 (0.004) | 0.497 (0.004) | | 0.423 (0.004) |
| IORBO + Prop. | ++ | ++ | ++ | | 0.600 (0.004) | 0.582 (0.004) | 0.577 (0.004) | |

# G  HYPER-PARAMETER SEARCH FOR ML ALGORITHMS

Figure 4 shows the hyper-parameter search process for an ML algorithm $\mathcal{M}_p$ on dataset $\mathcal{D}_j$. The optimal hyper-parameters, $\Theta^*_{\mathcal{M}_p, \mathcal{D}_j}$, were determined using five-fold cross-validation based on ML evaluation metrics.

# H  HYPER-PARAMETER SEARCH SPACES

Table 11: Logistic Regression search space for classification dataset.

| Parameter | Distribution |
|---|---|
| C | LogUniform $(-4, 4)$ |
| max_iter | IntUniform $(50, 200)$ |
| l1_ratio | Uniform $(0, 1)$ |
| algorithm | {"svd", "eig", "qr", "svd-qr", "svd-jacobi"} |
| solver | {"newton-cg", "lbfgs", "liblinear", "sag", "saga"} |
| class_weight | {"balanced", None} |
| number of tuning iterations | 30 |

For hyper-parameter search related to `TabDDPM`, please refer to the work by Kotelnikov et al. (2023).

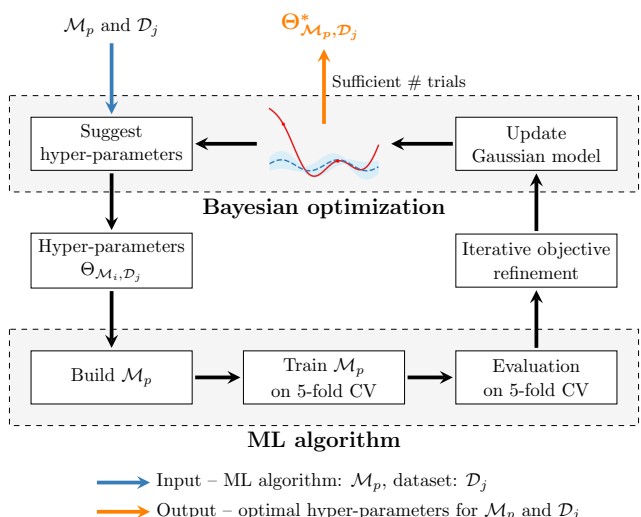

Figure 4: Hyper-parameter search for an ML algorithm.

Table 12: ElasticNet search space for regression dataset.

| Parameter | Distribution |
|---|---|
| alpha | Uniform $(1, 10)$ |
| max_iter | IntUniform $(100, 2000)$ |
| l1_ratio | Uniform $(0, 1)$ |
| tol | LogUniform $(10^{-5}, 10^{-1})$ |
| fit_intercept | {True, False} |
| normalize | {True, False} |
| number of tuning iterations | 30 |

Table 13: Bagging for Logistic Regression search space for classification dataset.

| Parameter | Distribution |
|---|---|
| C | LogUniform $(-4, 4)$ |
| max_iter | IntUniform $(50, 200)$ |
| l1_ratio | Uniform $(0, 1)$ |
| algorithm | {"svd", "eig", "qr", "svd-qr", "svd-jacobi"} |
| solver | {"qn"} |
| class_weight | {"balanced", None} |
| number of tuning iterations | 30 |

Table 14: Bagging for ElasticNet search space for regression dataset.

| Parameter | Distribution |
|---|---|
| alpha | Uniform $(1, 10)$ |
| max_iter | IntUniform $(100, 2000)$ |
| l1_ratio | Uniform $(0, 1)$ |
| tol | LogUniform $(10^{-5}, 10^{-1})$ |
| fit_intercept | {True, False} |
| normalize | {True, False} |
| number of tuning iterations | 30 |

Table 15: SVM search space for classification dataset (LinearSVC).

| Parameter | Distribution |
|---|---|
| C | LogUniform $(0.1, 10)$ |
| max_iter | IntUniform $(100, 1500)$ |
| tol | LogUniform $(-5, -1)$ |
| penalty | {"hinge", "squared_hinge"} |
| loss | {True, False} |
| fit_intercept | {True, False} |
| penalized_intercept | {True, False} |
| class_weight | {"balanced", None} |
| number of tuning iterations | 30 |

Table 16: SVM search space for regression dataset (LinearSVR).

| Parameter | Distribution |
|---|---|
| C | LogUniform $(0.1, 10)$ |
| max_iter | IntUniform $(100, 1500)$ |
| tol | LogUniform $(-5, -1)$ |
| epsilon | Uniform $(0, 1)$ |
| fit_intercept | {True, False} |
| penalized_intercept | {True, False} |
| number of tuning iterations | 30 |

Table 17: RF search space for classification and regression dataset (RandomForestClassifier and RandomForestRegressor).

| Parameter | Distribution |
|---|---|
| n_estimators | IntUniform $(50, 500)$ |
| max_depth | IntUniform $(10, 100)$ |
| min_samples_split | IntUniform $(2, 20)$ |
| min_samples_leaf | IntUniform $(1, 20)$ |
| max_features | {"sqrt", "log2"} |
| number of tuning iterations | 30 |

Table 18: XGBoost search space for classification and regression dataset (XGBClassifier and XGBRegressor).

| Parameter | Distribution |
|---|---|
| n_estimators | IntUniform $(50, 500)$ |
| max_depth | IntUniform $(3, 15)$ |
| learning_rate | Uniform $(0.01, 0.3)$ |
| subsample | Uniform $(0.5, 1)$ |
| colsample_bytree | Uniform $(0.5, 1)$ |
| gamma | Uniform $(0, 5)$ |
| reg_alpha | Uniform $(0, 1)$ |
| reg_lambda | Uniform $(0, 1)$ |
| scale_pos_weight | Uniform $(1, 10)$ |
| number of tuning iterations | 30 |

Table 19: `CTGAN`, `CopulaGAN` and `DP-CGANS` search space.

| Parameter | Distribution |
|---|---|
| epochs | IntUniform $(100, 2\,000, 100)$ |
| batch_size | IntUniform $(500, 30\,000, 100)$ |
| embedding_dim | $\{32, 64, 128, 256\}$ |
| generator_dim | $\{32, 64, 128, 256\}$ |
| discriminator_dim | $\{32, 64, 128, 256\}$ |
| generator_learning_rate | Uniform $(10^{-5}, 10^{-3})$ |
| generator_decay | Uniform $(10^{-7}, 10^{-5})$ |
| discriminator_learning_rate | Uniform $(10^{-5}, 10^{-3})$ |
| discriminator_decay | Uniform $(10^{-7}, 10^{-5})$ |
| $\alpha$ | Uniform $(10^{-2}, 10^{4})$ |
| $\beta$ | Uniform $(10^{-10}, 10^{1})$ |
| number of moments | $\{1,2,3,4\}$ |
| number of tuning iterations | 30 |

Table 20: `TVAE` search space.

| Parameter | Distribution |
|---|---|
| epochs | IntUniform $(100, 2\,000, 100)$ |
| batch_size | IntUniform $(500, 30\,000, 100)$ |
| embedding_dim | $\{32, 64, 128, 256\}$ |
| compress_dims | $\{32, 64, 128, 256\}$ |
| decompress_dim | $\{32, 64, 128, 256\}$ |
| loss_factor | $\{0.25, 0.5, 1, 2, 4\}$ |
| l2scale | Uniform $(10^{-6}, 10^{-4})$ |
| $\alpha$ | Uniform $(10^{-2}, 10^{4})$ |
| $\beta$ | Uniform $(10^{-10}, 10^{1})$ |
| number of moments | $\{1,2,3,4\}$ |
| number of tuning iterations | 30 |

Table 21: `CTAB-GAN` search space.

| Parameter | Distribution |
|---|---|
| epochs | IntUniform $(100, 2\,000, 100)$ |
| batch_size | IntUniform $(500, 4\,000, 100)$ |
| test_ratio | $\{0.1, 0.2, 0.3, 0.4, 0.5\}$ |
| n_class_layer | $\{1, 2, 3, 4\}$ |
| class_dim | $\{32, 64, 128, 256\}$ |
| random_dim | $\{16, 32, 64, 128\}$ |
| num_channels | $\{16, 32, 64\}$ |
| $\alpha$ | Uniform $(10^{-2}, 10^{4})$ |
| $\beta$ | Uniform $(10^{-10}, 10^{1})$ |
| number of moments | $\{1,2,3,4\}$ |
| number of tuning iterations | 30 |

