# OpenReview forum: "Improving Tabular Generative Models: Loss Functions, Benchmarks, and Iterative Objective Bayesian Approaches"
_ICLR.cc/2025/Conference — Submitted to ICLR 2025_

### Official Review · Reviewer_HdF4 · 2024-10-28

**Soundness:** 4
**Presentation:** 3
**Contribution:** 3
**Rating:** 6
**Confidence:** 4

**Summary:**

This work introduces correlation and moment-matching loss functions to regularize the loss function of different deep generative models for tabular data. Its results show that with proper selection of hyperparameters, its approach consistently improves the baselines. A Bayesian optimization procedure is introduced for hyperparameter tuning.

**Strengths:**

The added regularizers are described clearly and intuitively, with a well-defined methodology and comprehensive benchmark design. This approach encompasses various generative models and employs Bayesian optimization to identify optimal hyperparameter configurations. Consistent improvements over baseline models are demonstrated.

**Weaknesses:**

What I miss from the paper is a discussion on how to tune the method in the case of data heterogeneity and its performance and robustness in missing data scenarios.  How do the regularizers formulate in the case of counting distributions (e.g., Poisson likelihood) or ordinal variables? Do they consistently improve the results in the case of large fractions of missing entries in the database? I set my score to 6 since I feel that without a proper discussion on these aspects, the impact of the paper is limited.

**Questions:**

See above

---

> ### Author Response · Authors · 2024-11-18
> **Reply to Reviewer HdF4 [1]**
>
> > **What I miss from the paper is a discussion on how to tune the method in the case of data heterogeneity and its performance and robustness in missing data scenarios. How do the regularizers formulate in the case of counting distributions (e.g., Poisson likelihood) or ordinal variables?**
>
> We appreciate the reviewer’s valuable questions. Regarding data heterogeneity, we agree that handling diverse data types, such as counting distributions and ordinal variables, is important. Currently, our method assumes a homogeneous distribution of the data, as all the datasets we have evaluated exhibit homogeneous distributions.
>
> However, extending the regularizers to accommodate Poisson-like distributions is straightforward. For count-based variables, we can introduce an additional loss term to incorporate the Poisson likelihood. The Poisson log-likelihood, as demonstrated in [PyTorch’s PoissonNLLLoss](https://pytorch.org/docs/stable/generated/torch.nn.PoissonNLLLoss.html), can be used for these variables.
>
> For ordinal variables, there are two potential approaches: (1) a mean squared error (MSE)-based loss, treating the ordinal variable as a continuous approximation, or (2) the use of ordinal cross-entropy loss, which better captures the ordinal nature of the variables.
>
> Including these terms in our proposed loss function is simple and straightforward and can be tailored to various data types to improve its applicability to heterogeneous datasets.
>
> We have added the following sentence to the end of Distribution-aware loss function:
>
>  > "Additionally, incorporating loss terms for Poisson log-likelihood and ordinal variables into our method is straightforward and can be easily adapted to handle diverse data types."
>
>
> > **Do they consistently improve the results in the case of large fractions of missing entries in the database?**
>
> In our approach, we preprocess missing data by filling in the missing entries with a specific placeholder value (often -1) to signal the absence of data. The generative model (GM) is designed to handle this representation of missingness during training. Our correlation- and distribution-aware loss function leverages the available data, focusing on the relationships and distributions within observed data points. This enables the model to learn from incomplete datasets effectively, even with large fractions of missing values.
>
> In addition to your concerns, we would like to highlight that we conducted an additional experiment post-deadline to evaluate IORBO against SBO using both mean and median aggregation methods. The statistical tests revealed that IORBO consistently outperformed SBO in both aggregation methods. The Nemenyi post-hoc test and win rates (e.g., IORBO achieved 0.591 and 0.561 win rates over SBO-Mean and SBO-Median, respectively) demonstrate IORBO's robustness in handling multiple metrics across different units. This highlights IORBO’s effectiveness in optimizing diverse objectives without requiring theoretical convergence guarantees.

---

> > ### Author Response · Authors · 2024-11-18
> > **Reply to Reviewer HdF4 [3]**
> >
> > [continued from previous comment]
> >
> >  4. **Table 5**
> >    - *Previous Version*:
> >      > Table 5: Results of the Nemenyi post-hoc test and win rate (with standard error in parentheses) comparing the row to column method. For details on significance levels, refer to Table 1.
> >
> > | **BO Method**        |             | **IOR**            | **SBO-Mean**       | **SBO-Median**      |             | **IOR**            | **SBO-Mean**       | **SBO-Median**      |
> > |----------------------|-------------|--------------------|--------------------|---------------------|-------------|--------------------|--------------------|---------------------|
> > |                      |             | **Statistical Tests** |                    |                     |             | **Win Rate**       |                    |                     |
> > |                      |             | IOR                | SBO-Mean           | SBO-Median          |             | IOR                | SBO-Mean           | SBO-Median          |
> > | **IOR**              |             |                    | 0                  | ++                  |             | **0.527 (0.010)**   | **0.534 (0.010)**   |                     |
> > | **SBO-Mean**         |             | 0                  |                    | ++                  |             | 0.473 (0.010)      |                    | **0.543 (0.010)**   |
> > | **SBO-Median**       |             | --                 | --                 |                     |             | 0.466 (0.010)      | 0.457 (0.010)      |                     |
> >
> >    - *Updated Version*:
> >      > Table 5: Results of the Nemenyi post-hoc test and win rate (with standard error in parentheses) comparing the row to column method. For details on significance levels, refer to Table 1.
> >
> > | **BO Method**         |             | **Statistical Tests** |           |           |             | **Win Rate**        |           |           |
> > |-----------------------|-------------|-----------------------|-----------|-----------|-------------|---------------------|-----------|-----------|
> > |                       |             | **IOR**              | **SBO-Mean** | **SBO-Median** |             | **IOR**            | **SBO-Mean** | **SBO-Median** |
> > |                       |             |                       |           |           |             |                     |           |           |
> > | **IOR**               |             |                       | ++        | ++        |             | **0.591 (0.004)**   | **0.561 (0.004)** |           |
> > | **SBO-Mean**          |             | --                    |           | --        |             | 0.409 (0.004)       |           | 0.461 (0.004) |
> > | **SBO-Median**        |             | --                    | ++        |           |             | 0.439 (0.004)       | **0.539 (0.004)** |           |
> >
> > **Notes**:
> > - `++` indicates the row method is significantly better than the column method.
> > - `--` indicates the row method is significantly worse than the column method.

---

> > ### Comment · Reviewer_HdF4 · 2024-11-18
> >
> > Thank you for your detailed answers. I understand that adapting the regularization term to different data types is relatively straightforward. However, I am concerned about the practical challenges when using multiple versions of the regularizer—one for each data type. This approach likely introduces significantly varying ranges and may require extensive fine-tuning to balance them effectively. There is also a risk that one data type could dominate others, undermining the overall performance.
> >
> > Without experimental validation demonstrating how these challenges are addressed—e.g., strategies for balancing the terms or mitigating dominance—I find it difficult to assign a higher score to the paper. Including such experiments or an analysis would significantly strengthen the work.

---

> > > ### Author Response · Authors · 2024-11-25
> > > **Reply to Reviewer HdF4 [4]**
> > >
> > > > **Thank you for your detailed answers. I understand that adapting the regularization term to different data types is relatively straightforward. However, I am concerned about the practical challenges when using multiple versions of the regularizer—one for each data type. This approach likely introduces significantly varying ranges and may require extensive fine-tuning to balance them effectively. There is also a risk that one data type could dominate others, undermining the overall performance. Without experimental validation demonstrating how these challenges are addressed—e.g., strategies for balancing the terms or mitigating dominance—I find it difficult to assign a higher score to the paper. Including such experiments or an analysis would significantly strengthen the work.**
> > >
> > > Thank you for your thoughtful feedback. We appreciate your concerns and would like to address them thoroughly.
> > >
> > > In our earlier response, we stated:
> > >
> > >  > Currently, our method assumes a homogeneous distribution of the data, as all the datasets we have evaluated exhibit homogeneous distributions.
> > >
> > > However, upon reevaluating the twenty datasets used in our experiments, we identified several variables with ordinal or Poisson-like characteristics. Here is the updated breakdown:
> > >
> > >  - Adult: Education (ordinal)
> > >  - Buddy: Condition (ordinal)
> > >  - California housing: HouseAge (ordinal), Population (Poisson)
> > >  - Cardio: Cholesterol, Glucose (ordinal)
> > >  - Churn2: Tenure (ordinal)
> > >  - Diabetes/Diabetes Balanced: Diabetes_binary, GenHlth, MentHlth, PhysHlth, Education, Income (ordinal)
> > >  - House: bedrooms, bathrooms, stories (ordinal)
> > >  - Insuarance: children (ordinal)
> > >  - King: bedrooms, bathrooms, floors (ordinal)
> > >
> > > Currently, our approach does not differentiate among these variable types (e.g., ordinal, Poisson-like or non-ordered discrete). Instead, we use a unified distribution-aware loss to address continuous variables and all discrete variable types in the same manner. Despite this generalization, our results in Tables 2–4 show that the proposed loss function significantly enhances the fidelity of synthetic data generated by generative models (GMs). Furthermore, this improvement translates into better performance in downstream machine learning (ML) tasks, indicating that the loss function effectively handles diverse data types.
> > >
> > > By leveraging this unified loss function, we aim to mitigate imbalances that might arise from introducing separate regularization terms for different data types. This approach simplifies implementation and reduces the risk of certain terms dominating others, as might occur with multiple specialized regularizers.
> > >
> > > To clarify your concern, we added the following sentence under the "Distribution-aware loss function" section:
> > >
> > >  > By using a unified distribution-aware loss, we handle continuous and discrete variables in the same manner, simplifying implementation and preventing imbalances that could arise from separate regularization terms for different data types.
> > >
> > > Additionally, we would like to highlight the new-updated strong performance of our Iterative Objective Refinement Bayesian Optimization (IORBO). In our experiments, IORBO consistently outperformed standard Bayesian Optimization (SBO), achieving superior optimization results.
> > >
> > > We hope this explanation clarifies your concerns and provides additional context. Thank you again for your valuable input!

---

> > > > ### Comment · Reviewer_HdF4 · 2024-11-25
> > > >
> > > > Thank you again for your reply. Yes, your loss function is versatile enough to handle different data types—I missed that in my first read. One question, though: Since you have already worked with heterogeneous datasets in the databases you are using, can you evaluate the metrics separately for the different data types? The loss function can handle arbitrary data types, but does the optimization process prioritize one type (usually continuous) over another (such as discrete)?

---

> > > > > ### Author Response · Authors · 2024-11-27
> > > > > **Reply to Reviewer HdF4 [5]**
> > > > >
> > > > > > **Thank you again for your reply. Yes, your loss function is versatile enough to handle different data types—I missed that in my first read. One question, though: Since you have already worked with heterogeneous datasets in the databases you are using, can you evaluate the metrics separately for the different data types?**
> > > > >
> > > > > Thank you for your questions. As shown in Table 6 ("Description of Experimented Datasets"), we evaluated a diverse set of datasets, including: five datasets with only continuous variables, three datasets with only discrete variables, and twelve datasets with mixed variables. In Table 4, we present the evaluation of the metrics separately for these datasets. Based on these results, we can conclude that the proposed loss function performs effectively across all data and dataset types.
> > > > >
> > > > >
> > > > >
> > > > > > **The loss function can handle arbitrary data types, but does the optimization process prioritize one type (usually continuous) over another (such as discrete)?**
> > > > >
> > > > >
> > > > > The behavior of the optimization process can depend on several factors:
> > > > >
> > > > >  - The number of discrete vs. continuous variables in the dataset.
> > > > >  - The normalization or preprocessing methods used by the generative models (GMs) for discrete and continuous variables.
> > > > >
> > > > > For example, all the GMs we evaluated use one-hot encoding for discrete variables, while continuous variables are typically normalized using mode-specific methods. GANs and VAEs employ mode-specific normalization (MSN), which includes steps like variational Gaussian mixture models (VGM), probability computation, and normalization (Xu et al., 2019). In contrast, TabDDPM uses a quantile normalization for continuous variables. As a result, during training, GANs and VAEs produce tensor values typically ranging from -1 to 1, with scalars representing values within the mode and one-hot vectors indicating which mode of a Gaussian mixture is selected. For discrete variables, one-hot encoding represents values as 0 or 1.
> > > > >
> > > > > In TabDDPM, continuous and discrete features are treated differently as stated in our manuscript:
> > > > >
> > > > >  > Unlike other GMs, TabDDPM handles continuous and discrete features separately. For continuous features, TabDDPM predicts the Gaussian noise added through a forward Markov process. For discrete features, it predicts their one-hot encoded representation. To align our proposed loss functions with this characteristic, we adapted the correlation and distribution loss functions... to focus exclusively on discrete features. For continuous features, the Gaussian input noise is treated as the real data and the TabDDPM's predicted noise component as the synthetic data, incorporating a controlling parameter $\zeta$ into the $\mathcal{L}_{\text{distribution}}^{(c)}$ computation.
> > > > >
> > > > > Therefore, in TabDDPM, one-hot encoded values for discrete variables remain 0 or 1, while the predicted noise for **all** continuous variables should have values centered around 0 (with a mean of 0 and standard deviation of 1).
> > > > >
> > > > > Regarding the treatment of continuous and discrete variables during training, we treat all columns equally, regardless of whether they represent continuous or discrete variables. This means that each column, or class within a discrete variable (represented across multiple columns), is given equal weight to each column representing a continuous variable (which may be split across multiple columns).
> > > > >
> > > > > To answer your question directly:
> > > > >
> > > > >  - For datasets with only continuous variables: In GANs or VAEs, the optimization process does not prioritize any particular continuous variable. In TabDDPM, we also do not prioritize any specific continuous variable, as we treat all continuous variables collectively.
> > > > >  - For datasets with only discrete variables: In any GM, each class/category within each discrete variable will contribute equally to the optimization process.
> > > > >  - For datasets with mixed variables: In GANs or TVAE, each scalar representing values within a mode and one-hot encoded vectors for continuous variables, as well as each class/category in the discrete variables, will contribute equally to the optimization. In TabDDPM, both each class in the discrete variables and **all** continuous variables together receive equal weight. However, the controlling parameter for continuous variables is expected to be sufficiently large to balance their contribution during the optimization process.
> > > > >
> > > > >
> > > > > References:
> > > > >
> > > > >  1. Lei Xu, Maria Skoularidou, Alfredo Cuesta-Infante, and Kalyan Veeramachaneni. Modeling tabular data using conditional GAN. In Proceedings of the 33rd International Conference on Neural Information Processing Systems, pages 7335–7345, 2019.

---

> ### Author Response · Authors · 2024-11-18
> **Reply to Reviewer HdF4 [2]**
>
> Additions to the paper that we have made:
>
> 1. **Abstract:**
>    - *Previous Version*:
>      > "Further, the proposed IORBO outperformed the SBO with mean aggregation in terms of win rate and outperformed the SBO with median aggregation overall."
>    - *Updated Version*:
>      > "The IORBO consistently outperformed SBO, yielding superior optimization results."
>
> 2. **Section 3.6 - Benchmarking Framework:**
>    - *Added*:
>      > "Bayesian Optimization Method. To compare the performance of the IORBO with the SBO using mean and median aggregation methods, we fine-tuned each GM on each dataset across different loss functions, employing three evaluated BO approaches. Statistical tests were then conducted to evaluate these methods."
>
> 3. **Results Section:**
>    - *Previous Version*:
>      > "Bayesian optimization method. The performance of the IORBO was compared to the SBO using two aggregation methods (mean and median aggregation). The ML methods (Figure 1 and Step 2 and 3) were fine-tuned for each dataset using five-fold cross-validation on the ML evaluation metrics using different BO methods. The statistical tests were then employed to evaluate the three BO methods. Table 6 shows the results of the Nemenyi post-hoc test and win rate (with standard error in parentheses) comparing methods in the rows to those in the columns. The Nemenyi post-hoc test indicates that there is no significant difference between IORBO and SBO-Mean, but in terms of the win rate, the IORBO performs significantly better than the SBO-Mean with a win rate of 0.527. The IORBO method demonstrates significant improvement compared to the SBO-Median method, both in terms of the Nemenyi post-hoc test ($++$) and in terms of the win rate (0.534)."
>    - *Updated Version*:
>      > "Bayesian Optimization Method. The performance of the IORBO was compared to the SBO using two aggregation methods (mean and median aggregation). We fine-tuned each GM on each dataset across two loss functions, employing three evaluated BO approaches. Table 6 shows the results of the Nemenyi post-hoc test and win rate (with standard error in parentheses) comparing methods in the rows to those in the columns. The Nemenyi post-hoc test indicates that the IORBO is significantly better than the SBO-Mean and SBO-Median with win rates of 0.591 and 0.561, respectively. The results demonstrate that IORBO is robust in handling metrics with different units and its potential as a reliable, broadly applicable BO method."

---

### Official Review · Reviewer_mC61 · 2024-10-31

**Soundness:** 2
**Presentation:** 3
**Contribution:** 3
**Rating:** 5
**Confidence:** 3

**Summary:**

**Review of "Improving Tabular Generative Models: Loss Functions, Benchmarks, and Iterative Objective Bayesian Approaches"**

This paper proposes several methods to enhance deep generative models (DGMs) for synthetic data generation with a particular focus on tabular data. While the work presents promising results in experiment, certain aspects need further clarification and further improvement.

**Strengths:**

The paper presents an approach to enhancing Deep Generative Models (DGMs) for synthetic data generation on tabular data. Introduction of a correlation- and distribution-aware loss function, iterative objective refinement Bayesian optimization, and a detailed benchmarking framework are presented.

**Weaknesses:**

1. **"Moment Generating Function (MGF)":**
The term "Moment Generating Function (MGF)" appears to be misused. The paper discusses empirical moments themselves rather than the empirical MGF $\hat{M_X}(t)$ from which the $n$-th moments can be obtained by taking $n$-th derivatives wrt $t$ at $t=0$. [See *Casella, Statistical Inference, 1990* (pp61)]

2. **Biased Estimator in Synthetic Data:**
 A biased estimator is used to calculate the standard deviation. This includes the estimator on synthetic data sampled at size $B$, which is not enough for the biased estimator to converge to the unbiased one. It would be beneficial for the paper to address or justify this choice.

3. **Hyperparameter $\lambda$:**
Hyperparameter $\lambda$ in Eq.6 scales the $L_{\text{distribution}}$ in a manner the same as $\beta$ in custom losses, since $\lambda$ is
 proportional to  $L_{\text{distribution}}$ in Eq.6.  Simultaneous inclusion of $\lambda$ and $\beta$ in the hyperparameter search may lead to issues such as multi-collinearity for Bayesian optimization.

**Questions:**

1. **Significance Levels and Decision-Making:**
   In Table 1, the column for significance levels presents $p$-value ranges. A more detailed description of the decision based on the test statistic (or $p$-value obtained) may be helpful in understanding the experiment since a two-sided test is concerned.

2. **Distribution matching loss:**
It is possible for non-converging distributions to have similar moments, especially in lower orders. And, moment estimators of higher order moments introduce instability in the finite sample sense, and this instability goes up when the moment order goes up. It would be helpful if the author could justify using moments for distribution rather than the usual distance/score-based metrics for distribution similarity.



**Some Suggestions:**
 ***Reordering Loss Components:***
  For clearer presentation, consider swapping the order of the two proposed loss components to explain what $\mu$ and $\sigma$ are before presenting them in Eq.2.

---

> ### Author Response · Authors · 2024-11-18
> **Reply to Reviewer mC61 [1]**
>
> > **Moment Generating Function (MGF): The term "Moment Generating Function (MGF)" appears to be misused. The paper discusses empirical moments themselves rather than the empirical MGF, from which the $n$-th moments can be obtained by taking $n$-th derivatives wrt $t$ at $t=0$.**
>
> Thank you so much for pointing this out. This is a typo and we have fixed that.
>
>
> > **Biased Estimator in Synthetic Data: A biased estimator is used to calculate the standard deviation. This includes the estimator on synthetic data sampled at size $B$, which is not enough for the biased estimator to converge to the unbiased one. It would be beneficial for the paper to address or justify this choice.**
>
> Thank you for your questions. Regarding the use of a biased estimator for standard deviation (and moments), we clarify that during training, all moments are estimated on mini-batches. Mini-batch training is essential in deep learning to efficiently handle large datasets, reduce memory usage, and enable frequent model updates, despite introducing some bias (Masters & Luschi, 2018).
>
> Importantly, our correlation- and distribution-aware loss function demonstrates statistically significant improvements over vanilla loss functions, effectively capturing true data distributions and enhancing synthetic data quality.
>
> References:
>
>  1. Masters, Dominic, and Carlo Luschi. "Revisiting small batch training for deep neural networks." arXiv preprint arXiv:1804.07612 (2018).
>
>
>
> > **Hyper-parameter $\lambda$: hyper-parameter $\lambda$ in Eq.6 scales the $\mathcal{L}_{\text{distribution}}$ in a manner the same as $\beta$ in custom losses, since $\lambda$ is proportional to $\mathcal{L}_{\text{distribution}}$ in Eq.6. Simultaneous inclusion of $\alpha$ and $\beta$ in the hyper-parameter search may lead to issues such as multi-collinearity for Bayesian optimization.**
>
> In Equation 6, the hyper-parameter $\lambda$ is used to scale the distribution loss term $\mathcal{L}_{\text{distribution}}$, similarly to how $\beta$ is applied in custom loss functions. As you said, $\lambda$ is proportional to the distribution term which can affect how strongly the model focuses on capturing the distributional aspects of the data. While each moment (e.g., mean, variance) could theoretically have its own weight, in our implementation we chose to simplify by fixing $\lambda=1$ to avoid increasing the number of hyper-parameters that need to be fine-tuned and avoid the potential issue of multi-collinearity. This makes the model easier to optimize and reduces the complexity of the hyper-parameter search.
>
> We have updated our manuscript as below:
>
>  - *Previous Version*:
>  > Finally, the distribution loss was defined as... where the number of moments, $H$, and the regularization parameter, $\lambda$, were hyper-parameters. Instead of making the moments equal, their quotient was made to be equal to one as a way to handle scale differences.
>
>  - *Updated Version*:
>  > Finally, the distribution loss was defined as... where the number of moments, $H$, and the regularization parameter, $\lambda$, were hyper-parameters. Instead of making the moments equal, their quotient was made to be equal to one as a way to handle scale differences. To avoid increasing the number of hyper-parameters that need to be fine-tuned and the potential issue of multi-collinearity, we fixed $\lambda=1$.

---

> ### Author Response · Authors · 2024-11-18
> **Reply to Reviewer mC61 [2]**
>
> > **Significance Levels and Decision-Making: In Table 1, the column for significance levels presents $p$-value ranges. A more detailed description of the decision based on the test statistic (or $p$-value obtained) may be helpful in understanding the experiment since a two-sided test is concerned.**
>
> Thank you for your comment. We appreciate the suggestion and agree that providing a more detailed explanation of the significance levels and decision-making process would benefit the clarity of our results.
>
> In Table 1, the significance levels are based on the commonly accepted interpretation of $p$-values in hypothesis testing. A $p$-value less than or equal to 0.01 ($p \leq 0.01$) indicates that the result is *highly significant*, meaning that the null hypothesis can be rejected with high confidence. A $p$-value between 0.01 and 0.05 ($0.01 < p \leq 0.05$) indicates *significant* results, where there is still a reasonable level of evidence against the null hypothesis, though not as strong as for the highly significant results. For $p$-values greater than 0.05, we consider the result not to be significant, indicating insufficient evidence to reject the null hypothesis.
>
> Regarding the two-sided test, the Nemenyi post-hoc test used in our analysis is based on the Friedman test, which is a non-parametric test for repeated measures. The Nemenyi test performs pairwise comparisons between the groups following the Friedman test and is a two-sided test. This means that the test evaluates whether the differences between the groups are statistically significant in both directions, i.e., it considers whether one group is significantly better or worse than another group.
>
> For transparency, we have added this explanation to the Appendix to provide readers with further insights into our decision-making criteria. We appreciate your suggestion to clarify this and hope the updated version adds clarity.
>
>
> > **Distribution matching loss: It is possible for non-converging distributions to have similar moments, especially in lower orders. And, moment estimators of higher order moments introduce instability in the finite sample sense, and this instability goes up when the moment order goes up. It would be helpful if the author could justify using moments for distribution rather than the usual distance/score-based metrics for distribution similarity.**
>
> We recognize the potential limitations of using moments, especially higher-order moments. However, moments offer several advantages, particularly in terms of interpretability and computational efficiency when compared to distance-based metrics. Lower-order moments (e.g., mean, variance) are especially stable and provide robust alignment with the distribution characteristics observed in tabular data.
>
> Regarding the choice of moments over traditional distance metrics, such as Wasserstein or MMD, our method aims to reduce computational complexity while capturing essential distributional features. Distance-based metrics are effective but can be computationally intensive, particularly for high-dimensional data. Using moments allows us to approximate the distribution with fewer computations, maintaining model efficiency. Additionally, since our method operates on tabular data where moments are generally representative of the underlying distribution, we found that our approach achieved adequate accuracy without stability issues.
>
> We have added the following sentence to the end of Distribution-aware loss function:
>
>  > "The choice of moments over distance-based metrics, such as Wasserstein, is motivated by their computational efficiency and stability, as lower-order moments provide a robust approximation of the distribution while avoiding the high computational cost associated with distance-based methods."

---

> ### Author Response · Authors · 2024-11-18
> **Reply to Reviewer mC61 [3]**
>
> > **Some Suggestions: Reordering Loss Components: For clearer presentation, consider swapping the order of the two proposed loss components to explain what $\mu$ and $\sigma$ are before presenting them in Eq.2.**
>
> Thank you for your suggestions. After careful consideration, we have decided to retain the current order, as it better aligns with the flow of the manuscript.
>
> In addition to your concerns, we would like to highlight that we conducted an additional experiment post-deadline to evaluate IORBO against SBO using both mean and median aggregation methods. The statistical tests revealed that IORBO consistently outperformed SBO in both aggregation methods. The Nemenyi post-hoc test and win rates (e.g., IORBO achieved 0.591 and 0.561 win rates over SBO-Mean and SBO-Median, respectively) demonstrate IORBO's robustness in handling multiple metrics across different units. This highlights IORBO’s effectiveness in optimizing diverse objectives without requiring theoretical convergence guarantees.
>
> Additions to the paper that we have made:
>
> 1. **Abstract:**
>    - *Previous Version*:
>      > "Further, the proposed IORBO outperformed the SBO with mean aggregation in terms of win rate and outperformed the SBO with median aggregation overall."
>    - *Updated Version*:
>      > "The IORBO consistently outperformed SBO, yielding superior optimization results."
>
> 2. **Section 3.6 - Benchmarking Framework:**
>    - *Added*:
>      > "Bayesian Optimization Method. To compare the performance of the IORBO with the SBO using mean and median aggregation methods, we fine-tuned each GM on each dataset across different loss functions, employing three evaluated BO approaches. Statistical tests were then conducted to evaluate these methods."
>
> 3. **Results Section:**
>    - *Previous Version*:
>      > "Bayesian optimization method. The performance of the IORBO was compared to the SBO using two aggregation methods (mean and median aggregation). The ML methods (Figure 1 and Step 2 and 3) were fine-tuned for each dataset using five-fold cross-validation on the ML evaluation metrics using different BO methods. The statistical tests were then employed to evaluate the three BO methods. Table 6 shows the results of the Nemenyi post-hoc test and win rate (with standard error in parentheses) comparing methods in the rows to those in the columns. The Nemenyi post-hoc test indicates that there is no significant difference between IORBO and SBO-Mean, but in terms of the win rate, the IORBO performs significantly better than the SBO-Mean with a win rate of 0.527. The IORBO method demonstrates significant improvement compared to the SBO-Median method, both in terms of the Nemenyi post-hoc test ($++$) and in terms of the win rate (0.534)."
>    - *Updated Version*:
>      > "Bayesian Optimization Method. The performance of the IORBO was compared to the SBO using two aggregation methods (mean and median aggregation). We fine-tuned each GM on each dataset across two loss functions, employing three evaluated BO approaches. Table 6 shows the results of the Nemenyi post-hoc test and win rate (with standard error in parentheses) comparing methods in the rows to those in the columns. The Nemenyi post-hoc test indicates that the IORBO is significantly better than the SBO-Mean and SBO-Median with win rates of 0.591 and 0.561, respectively. The results demonstrate that IORBO is robust in handling metrics with different units and its potential as a reliable, broadly applicable BO method."

---

> ### Author Response · Authors · 2024-11-18
> **Reply to Reviewer mC61 [4]**
>
> [continued from previous comment]
>
> 4. **Table 5**
>    - *Previous Version*:
>      > Table 5: Results of the Nemenyi post-hoc test and win rate (with standard error in parentheses) comparing the row to column method. For details on significance levels, refer to Table 1.
>
> | **BO Method**        |             | **IOR**            | **SBO-Mean**       | **SBO-Median**      |             | **IOR**            | **SBO-Mean**       | **SBO-Median**      |
> |----------------------|-------------|--------------------|--------------------|---------------------|-------------|--------------------|--------------------|---------------------|
> |                      |             | **Statistical Tests** |                    |                     |             | **Win Rate**       |                    |                     |
> |                      |             | IOR                | SBO-Mean           | SBO-Median          |             | IOR                | SBO-Mean           | SBO-Median          |
> | **IOR**              |             |                    | 0                  | ++                  |             | **0.527 (0.010)**   | **0.534 (0.010)**   |                     |
> | **SBO-Mean**         |             | 0                  |                    | ++                  |             | 0.473 (0.010)      |                    | **0.543 (0.010)**   |
> | **SBO-Median**       |             | --                 | --                 |                     |             | 0.466 (0.010)      | 0.457 (0.010)      |                     |
>
>    - *Updated Version*:
>      > Table 5: Results of the Nemenyi post-hoc test and win rate (with standard error in parentheses) comparing the row to column method. For details on significance levels, refer to Table 1.
>
> | **BO Method**         |             | **Statistical Tests** |           |           |             | **Win Rate**        |           |           |
> |-----------------------|-------------|-----------------------|-----------|-----------|-------------|---------------------|-----------|-----------|
> |                       |             | **IOR**              | **SBO-Mean** | **SBO-Median** |             | **IOR**            | **SBO-Mean** | **SBO-Median** |
> |                       |             |                       |           |           |             |                     |           |           |
> | **IOR**               |             |                       | ++        | ++        |             | **0.591 (0.004)**   | **0.561 (0.004)** |           |
> | **SBO-Mean**          |             | --                    |           | --        |             | 0.409 (0.004)       |           | 0.461 (0.004) |
> | **SBO-Median**        |             | --                    | ++        |           |             | 0.439 (0.004)       | **0.539 (0.004)** |           |
>
> **Notes**:
> - `++` indicates the row method is significantly better than the column method.
> - `--` indicates the row method is significantly worse than the column method.

---

> > ### Comment · Reviewer_mC61 · 2024-11-22
> > **Response to Rebuttal**
> >
> > Highly appreciate your effort in making a detailed response.
> >
> > 1.  `we chose to simplify by fixing $\lambda=1$
> >  to avoid increasing the number of hyper-parameters that need to be fine-tuned and avoid the potential issue of multi-collinearity. This makes the model easier to optimize and reduces the complexity of the hyper-parameter search.`
> > The author proposes to fix hyperparameter $\lambda$ that was previously used to feed into the BO. There should be an update on the experiment results. This can also serve as a check for the validity of the assessment framework, i.e., how the metrics behave after dealing with the possible multi-collinearity in BO.
> > Additionally, $\lambda$ should no longer be called "hyperparameter" if you fix the value. And since $\lambda$ is not mentioned in the manuscript after introducing it in Eq.6, I'll suggest removing it from the manuscript.
> >
> > 2. I'm not convinced that having similar moments means distributions are similar. There could be different distributions having similar first few moments.
> > This is my biggest concern that prevents me from adjusting my rating. The author is encouraged to present proof to claim the validity of the approach. I do see that the author tries to claim the empirical effectiveness of the proposed approach in their response to reviewer 1iCq, and I would be convinced it if the author could show the empirical validity through a well-crafted experiment. Not from the general performance improvement, but something specific to showcase this.
> >
> > 3. In Table 1, the author is reminded again to look at the column where the header presents "Significance Level" (typically refer to the Type I error rate $\alpha$) but shows ranges of $p$-values.
> >
> > I'll be open to adjusting my rating if the author addresses my concerns.

---

> > > ### Author Response · Authors · 2024-11-25
> > > **Reply to Reviewer mC61 [5]**
> > >
> > > > **We chose to simplify by fixing $\lambda=1$ to avoid increasing the number of hyper-parameters that need to be fine-tuned and avoid the potential issue of multi-collinearity. This makes the model easier to optimize and reduces the complexity of the hyper-parameter search. The author proposes to fix hyperparameter $\lambda$ that was previously used to feed into the BO. There should be an update on the experiment results. This can also serve as a check for the validity of the assessment framework, i.e., how the metrics behave after dealing with the possible multi-collinearity in BO. Additionally, $\lambda$ should no longer be called "hyperparameter" if you fix the value. And since $\lambda$ is not mentioned in the manuscript after introducing it in Eq.6, I'll suggest removing it from the manuscript.**
> > >
> > > Thank you for your questions. To clarify, we set the value of $\lambda = 1$ prior to running the experiments, and all results presented are based on this fixed value. We fully acknowledge that $\lambda$ should no longer be considered a "hyperparameter." As such, we have revised Eq. 6 and updated the accompanying text below Eq. 6 as follows:
> > >
> > >  - *Previous Version*:
> > >  > "...where the number of moments, $H$, and the regularization parameter, $\lambda$, were hyper-parameters."
> > >  - *Updated Version*:
> > >  > "...where the number of moments, $H$, was hyper-parameter."
> > >
> > >
> > >
> > > > **I'm not convinced that having similar moments means distributions are similar. There could be different distributions having similar first few moments. This is my biggest concern that prevents me from adjusting my rating. The author is encouraged to present proof to claim the validity of the approach. I do see that the author tries to claim the empirical effectiveness of the proposed approach in their response to reviewer 1iCq, and I would be convinced it if the author could show the empirical validity through a well-crafted experiment. Not from the general performance improvement, but something specific to showcase this.**
> > >
> > > The method we use is rooted in a solid statistical foundation: The method of moments.
> > >
> > > The method of moments (introduced by Pearson in 1936) is a consistent estimator, meaning that under certain assumptions, the estimated parameters converge to the true parameters as the sample size becomes sufficiently large (Rice, 2007; Silvey, 2017). In our case, we augment the maximum likelihood estimation (MLE), which is performed during the training of the models, with the method of moments. While the method of moments is traditionally an alternative to MLE, combining it with MLE ensures that the model aligns with the true distribution both in terms of its moments (statistical properties like mean, variance, etc.) and the likelihood function. This dual approach reinforces the MLE-based estimation by explicitly incorporating key statistical properties of the data distribution.
> > >
> > > To clarify and validate this approach, we added the following explanation under the "Distribution-aware loss function" section:
> > >
> > >  > The distribution-aware loss function integrates the strengths of the method of moments and maximum likelihood estimation (MLE) to align with the true distribution by capturing both statistical moments and likelihood properties. This integration enhances the model's ability to learn accurate data representations (Pearson, 1936; Rice, 2007).
> > >
> > >
> > > References:
> > >
> > >  1. Pearson, Karl. "Method of moments and method of maximum likelihood." Biometrika 28, no. 1/2 (1936): 34-59.
> > >  2. Rice, John A., and John A. Rice. Mathematical statistics and data analysis. Vol. 371. Belmont, CA: Thomson/Brooks/Cole, 2007.
> > >  3. Silvey, Samuel David. Statistical inference. Routledge, 2017.
> > >
> > >
> > >
> > > > **In Table 1, the author is reminded again to look at the column where the header presents "Significance Level" (typically refer to the Type I error rate $\alpha$) but shows ranges of $p$-values.**
> > >
> > > Thank you so much for your reminder. We have updated our manuscript accordingly. Now, we use "$p$-value ranges" instead.

---

> > > > ### Comment · Reviewer_mC61 · 2024-11-26
> > > > **Response to the Author**
> > > >
> > > > Thank you very much for the response!
> > > >
> > > > The author proposes to claim the effectiveness of the distribution matching by empirical moments using arguments from methods of moments. At the intuition level, it works. I see the logic behind it and have adjusted my rating.
> > > >
> > > > However, a crucial gap remains as there is no theoretical proof supporting the effectiveness of this method.
> > > > For instance, consider the density was a mixture of two normal density functions (which are common in data if a bi-heterogeneity is present) with 5 unknown parameters $\lambda, \mu_1, \mu_2, \sigma_1, \sigma_2$.
> > > > Since BO for the number of moments, $H$, ranges from 1 to 4 in the experiments, these moments may not fully represent the distribution. There's also the issue of potential numerical instability when incorporating more moments.
> > > > The absence of a theoretical guarantee is accompanied by the absence of relevant experiment results to demonstrate this idea, which makes me hard to fully endorse this method of distribution matching. It would strengthen the credibility to dive deeper into what they claim in the response "Lower-order moments (e.g., mean, variance) ... provide robust alignment with the distribution characteristics observed in tabular data.", and such lower-order moments are (asymptotically, or empirically) sufficient.
> > > >
> > > > Lastly, the text in lines 160-168 could be clearer. After several revisions, it has become somewhat unorganized and dense.
> > > >
> > > > I am still flexible in adjusting my rating.

---

> > > > > ### Author Response · Authors · 2024-11-27
> > > > > **Reply to Reviewer mC61 [6]**
> > > > >
> > > > > > **However, a crucial gap remains as there is no theoretical proof supporting the effectiveness of this method. For instance, consider the density was a mixture of two normal density functions (which are common in data if a bi-heterogeneity is present) with 5 unknown parameters $\lambda$, $\mu_1$, $\mu_2$, $\sigma_1$, $\sigma_2$. Since BO for the number of moments, $H$, ranges from 1 to 4 in the experiments, these moments may not fully represent the distribution. There's also the issue of potential numerical instability when incorporating more moments. The absence of a theoretical guarantee is accompanied by the absence of relevant experiment results to demonstrate this idea, which makes me hard to fully endorse this method of distribution matching. It would strengthen the credibility to dive deeper into what they claim in the response "Lower-order moments (e.g., mean, variance) ... provide robust alignment with the distribution characteristics observed in tabular data.", and such lower-order moments are (asymptotically, or empirically) sufficient.**
> > > > >
> > > > > Thank you for your insightful question.
> > > > >
> > > > > While we acknowledge the importance of theoretical guarantees, we do not currently have a formal proof supporting the effectiveness of this method. We can offer some intuitive reasoning: The loss can be thought of as "pinning down" the low-frequency components of the distribution (i.e., the lower-order moments), which stabilizes the GM's learning process by ensuring that it does not deviate significantly from the observed distribution in these aspects. By constraining the first few moments, the GM is regularized, and can "focus" on matching the higher-frequency (higher moments) characteristics of the real data distribution.
> > > > >
> > > > > This principle bears some resemblance to gradient boosting models: The initial models in gradient boosting tend to approximate the dominant low-frequency patterns in the data, while subsequent models iteratively refine the residual, capturing higher-frequency information. Analogously, in our approach, the moment-matching loss captures the lower-order moments, while the GM implicitly learns the residual higher-order characteristics of the data distribution. The presented results indicate that the proposed loss significantly reduces the variance of the distribution estimation.
> > > > >
> > > > > Regarding the specific concern raised about multimodal distributions (e.g., a mixture of two Gaussian densities), we believe there may be a potential misinterpretation. The distribution-aware loss does not need to fully represent the data distribution in its entirety. Instead, the GM learns to account for characteristics of the distribution that are not explicitly encoded in the moment constraints. For instance, while the proposed loss explicitly aligns the first four moments in the experiments, the GM itself learns to approximate other features of the multimodal distribution, such as the positions and relative weights of distinct modes. Our results demonstrate that this approach is robust and effective across a variety of tabular data distributions.
> > > > >
> > > > > Concerning the issue of higher-order moments, while we can't provide theoretical guarantees, the empirical results suggest that the lower-order moments (up to the 4th moment in the experiments) capture critical distributional characteristics of tabular data.
> > > > >
> > > > > Regarding your comment on potential numerical instability when incorporating more moments, we agree that this is a valid concern. One possible solution could involve using an exponential moving average of the moments over iterations to ensure the moments match on average, rather than for a single mini-batch. In response, we have added this as a potential future improvement in the conclusion of the paper.
> > > > >
> > > > > We appreciate your feedback and hope this explanation clarifies the rationale behind our approach.
> > > > >
> > > > >
> > > > >
> > > > > > **Lastly, the text in lines 160-168 could be clearer. After several revisions, it has become somewhat unorganized and dense.**
> > > > >
> > > > > Thank you so much for your suggestion.  We completely agree with your feedback. To improve clarity, we have restructured the section by moving the two sentences from lines 160-168 to the beginning of the "Distribution-aware loss function" section, where they now serve as the motivations for the approach.

---

> > > > > > ### Comment · Reviewer_mC61 · 2024-12-03
> > > > > > **Response to author**
> > > > > >
> > > > > > I appreciate the authors' thorough response - they are helpful to better understand the claim regarding the distribution matching loss at the intuition level. However, there is no specific experiment that demonstrates the effectiveness of the proposed $L_{\text{distribution}}$ (1. The empirical results are from a hodgepodge of newly proposed methods, including the introduction of $L_\text{correlation}$, $L_{\text{distribution}}$, and various improvements from SBO, not from $L_{\text{distribution}}$ alone. 2. I do not see how the claim "the empirical results suggest that the lower-order moments (up to the 4th moment in the experiments) capture critical distributional characteristics of tabular data." is shown in the empirical results). I believe the paper can benefit from e.g., providing some more empirical evidence to substantiate.

---

> > > > > > > ### Author Response · Authors · 2024-12-04
> > > > > > > **Response**
> > > > > > >
> > > > > > > Thank you for your answer. We actually performed ablation studies and prove that each component contributes.

---

### Official Review · Reviewer_XQkD · 2024-10-31

**Soundness:** 2
**Presentation:** 3
**Contribution:** 3
**Rating:** 5
**Confidence:** 4

**Summary:**

- Introduced a correlation- and distribution-aware loss function designed as a regularizer for DGMs in tabular data synthesis that displays promising results
- Introduced a hyperparameter tuning approach, IORBO, that leverages rank-based aggregation. (concerns of units
- They introduce a benchmarking system evaluating statistical similarity, ML TSTR performance, and ML augmentation performance, with robust statistical tests.

**Strengths:**

**Originality and Quality**

The correlation- and distribution-aware loss function is new and interesting to me. I have not encountered works that display the effectiveness of enforcing correlation and high-order moments in the loss function to improve generative models. It is nice to see an improvement in existing hyperparameter tuning algorithms such as Standard Bayesian Optimization by adding an iterative refinement process.

**Clarity**

Individual sections of the paper are well written.

**Significance**

Tabular data generation is gaining traction in real-world applications such as electronic health records. This work helps bring progress to tabular data generation.

**Weaknesses:**

- I am struggling to find a central theme/research question the paper is trying to answer. It provides solutions from three different perspectives: 1) Loss Function Regularization: Improving generative model outputs by enforcing statistical properties (e.g., correlation, distribution); 2) Hyperparameter Tuning: Using methods like IORBO for iterative optimization; 3) Statistical Tests: Providing a framework for assessing model performance across metrics. I am unable to determine a flow to link the three ideas together/how one idea enforces the other.
- L486: How does IORBA perform against other hyperparameter tuning methods such as [Randomised Optimization, GridSearch etc.](https://scikit-learn.org/1.5/modules/grid_search.html#tuning-the-hyper-parameters-of-an-estimator) in terms of performance? What about the computational cost for IORBA vs. SBO and other mentioned baselines, what is this tradeoff? Additionally, what are the optimized hyperparameters that you obtain from your method? Ablation studies of the aforementioned would make your case stronger.
- In [TabSyn](https://arxiv.org/abs/2310.09656), the authors provided a comprehensive evaluation of synthetic tabular data using over five distinct evaluation metrics. Their metrics are straightforward and easy to comprehend. It will be nice to compare and justify why your metrics are more convincing and better than their proposed benchmark so that users should use your metrics instead of/in addition to TabSyn’s.
- Privacy is also crucial in synthetic tabular generation. How does your proposed loss function affect privacy-preserving metrics such as DCR and C2ST?

**Questions:**

The individual contributions of the paper are good. However, my main concern is the overall theme of the paper. I am unable to determine the overall research question the paper is trying to address. Please see weaknesses.

---

> ### Author Response · Authors · 2024-11-18
> **Reply to Reviewer XQkD [1]**
>
> > **I am struggling to find a central theme/research question the paper is trying to answer. It provides solutions from three different perspectives: 1) Loss Function Regularization: Improving generative model outputs by enforcing statistical properties (e.g., correlation, distribution); 2) hyper-parameter Tuning: Using methods like IORBO for iterative optimization; 3) Statistical Tests: Providing a framework for assessing model performance across metrics. I am unable to determine a flow to link the three ideas together/how one idea enforces the other.**
>
> Thank you for your questions. The central theme of our paper revolves around improving the quality of synthetic data generated by generative models (GMs), particularly for tabular data. The three components you mentioned—loss function regularization, hyper-parameter tuning, and statistical tests—are interconnected.  Each component addresses a critical challenge in synthetic data generation:
>
>  1. Loss Function Regularization: We introduce a correlation- and distribution-aware loss function to ensure that GMs capture the true underlying distribution and complex dependencies present in tabular data. This forms the foundation to improve the synthetic data's fidelity to real-world data.
>  2. Hyper-parameter Tuning (IORBO): To further optimize the generative process, we propose IORBO to select the best hyper-parameters. IORBO directly addresses the challenges of aggregating multiple metrics in Bayesian optimization (BO), enabling more meaningful comparisons and improving the overall optimization process.
>  3. Statistical Tests and Benchmarking: Finally, we provide a robust benchmarking framework to evaluate the performance of GMs and the synthetic data they produce. By using real-world datasets and established baselines, we demonstrate how our method leads to better model performance in downstream tasks.
>
> These three components work together by first improving the data generation process through regularization, then refining the model through better hyper-parameter optimization, and ultimately validating the improvements through rigorous statistical evaluation. We have rewritten the Abstract and the Introduction to clarify the research theme and contributions as follows:
>
> Abstract:
>
>  > "These challenges raise two key research questions: How can GMs be refined to capture the complexities of real-world data better? How can hyper-parameter optimization approaches be adapted to handle diverse evaluation metrics effectively? To address these gaps, we introduce a novel correlation- and distribution-aware loss function that regularizes GMs, enhancing their ability to generate synthetic tabular data that faithfully represents actual distributions. We also propose IORBO, which ranks metrics to enable clear comparisons across diverse objectives."
>
> Introduction:
>
>  > "This work centers on improving GM performance through more effective hyper-parameter tuning, enhanced data generation techniques, and comprehensive evaluation across diverse metrics, contributing with..."

---

> ### Author Response · Authors · 2024-11-18
> **Reply to Reviewer XQkD [2]**
>
> > **How does IORBA perform against other hyper-parameter tuning methods such as Randomised Optimization, GridSearch etc. in terms of performance?**
>
> We appreciate the suggestion to compare IORBA with other hyper-parameter tuning methods like Randomized Optimization and GridSearch. Here is why we focused on Bayesian Optimization (BO) for our comparison:
>
>  1. Efficiency: BO is more efficient than Randomized Optimization and GridSearch because it uses probabilistic models to intelligently explore the search space based on prior evaluations, leading to faster convergence and better performance with fewer evaluations (Snoek et al., 2012).
>  2. Theoretical support: BO reduces computational cost by using surrogate models (e.g., Gaussian Processes), which predict hyper-parameter performance, minimizing the need for exhaustive trials, as seen in GridSearch and Randomized Optimization (Bergstra et al., 2011).
>
> Given these advantages, we chose to focus on comparing Iterative Objective Refinement Bayesian Optimization (IORBO) with Standard Bayesian Optimization (SBO).
>
> To address the comparison, we conducted an additional experiment post-deadline to evaluate IORBO against SBO using both mean and median aggregation methods. We fine-tuned each GM across various datasets with different loss functions and compared the three BO approaches.
>
> The statistical tests revealed that IORBO consistently outperformed SBO in both aggregation methods. The Nemenyi post-hoc test and win rates (e.g., IORBO achieved 0.591 and 0.561 win rates over SBO-Mean and SBO-Median, respectively) demonstrate IORBO's robustness in handling multiple metrics across different units. This highlights IORBO’s effectiveness in optimizing diverse objectives without requiring theoretical convergence guarantees.
>
> Additions to the paper:
>
> 1. **Abstract:**
>    - *Previous Version*:
>      > "Further, the proposed IORBO outperformed the SBO with mean aggregation in terms of win rate and outperformed the SBO with median aggregation overall."
>    - *Updated Version*:
>      > "The IORBO consistently outperformed SBO, yielding superior optimization results."
>
> 2. **Section 3.6 - Benchmarking Framework:**
>    - *Added*:
>      > "Bayesian Optimization Method. To compare the performance of the IORBO with the SBO using mean and median aggregation methods, we fine-tuned each GM on each dataset across different loss functions, employing three evaluated BO approaches. Statistical tests were then conducted to evaluate these methods."
>
> 3. **Results Section:**
>    - *Previous Version*:
>      > "Bayesian optimization method. The performance of the IORBO was compared to the SBO using two aggregation methods (mean and median aggregation). The ML methods (Figure 1 and Step 2 and 3) were fine-tuned for each dataset using five-fold cross-validation on the ML evaluation metrics using different BO methods. The statistical tests were then employed to evaluate the three BO methods. Table 6 shows the results of the Nemenyi post-hoc test and win rate (with standard error in parentheses) comparing methods in the rows to those in the columns. The Nemenyi post-hoc test indicates that there is no significant difference between IORBO and SBO-Mean, but in terms of the win rate, the IORBO performs significantly better than the SBO-Mean with a win rate of 0.527. The IORBO method demonstrates significant improvement compared to the SBO-Median method, both in terms of the Nemenyi post-hoc test ($++$) and in terms of the win rate (0.534)."
>    - *Updated Version*:
>      > "Bayesian Optimization Method. The performance of the IORBO was compared to the SBO using two aggregation methods (mean and median aggregation). We fine-tuned each GM on each dataset across two loss functions, employing three evaluated BO approaches. Table 6 shows the results of the Nemenyi post-hoc test and win rate (with standard error in parentheses) comparing methods in the rows to those in the columns. The Nemenyi post-hoc test indicates that the IORBO is significantly better than the SBO-Mean and SBO-Median with win rates of 0.591 and 0.561, respectively. The results demonstrate that IORBO is robust in handling metrics with different units and its potential as a reliable, broadly applicable BO method."

---

> ### Author Response · Authors · 2024-11-18
> **Reply to Reviewer XQkD [3]**
>
> [continued from previous comment]
>
> 4. **Table 5**
>    - *Previous Version*:
>      > Table 5: Results of the Nemenyi post-hoc test and win rate (with standard error in parentheses) comparing the row to column method. For details on significance levels, refer to Table 1.
>
> | **BO Method**        |             | **IOR**            | **SBO-Mean**       | **SBO-Median**      |             | **IOR**            | **SBO-Mean**       | **SBO-Median**      |
> |----------------------|-------------|--------------------|--------------------|---------------------|-------------|--------------------|--------------------|---------------------|
> |                      |             | **Statistical Tests** |                    |                     |             | **Win Rate**       |                    |                     |
> |                      |             | IOR                | SBO-Mean           | SBO-Median          |             | IOR                | SBO-Mean           | SBO-Median          |
> | **IOR**              |             |                    | 0                  | ++                  |             | **0.527 (0.010)**   | **0.534 (0.010)**   |                     |
> | **SBO-Mean**         |             | 0                  |                    | ++                  |             | 0.473 (0.010)      |                    | **0.543 (0.010)**   |
> | **SBO-Median**       |             | --                 | --                 |                     |             | 0.466 (0.010)      | 0.457 (0.010)      |                     |
>
>    - *Updated Version*:
>      > Table 5: Results of the Nemenyi post-hoc test and win rate (with standard error in parentheses) comparing the row to column method. For details on significance levels, refer to Table 1.
>
> | **BO Method**         |             | **Statistical Tests** |           |           |             | **Win Rate**        |           |           |
> |-----------------------|-------------|-----------------------|-----------|-----------|-------------|---------------------|-----------|-----------|
> |                       |             | **IOR**              | **SBO-Mean** | **SBO-Median** |             | **IOR**            | **SBO-Mean** | **SBO-Median** |
> |                       |             |                       |           |           |             |                     |           |           |
> | **IOR**               |             |                       | ++        | ++        |             | **0.591 (0.004)**   | **0.561 (0.004)** |           |
> | **SBO-Mean**          |             | --                    |           | --        |             | 0.409 (0.004)       |           | 0.461 (0.004) |
> | **SBO-Median**        |             | --                    | ++        |           |             | 0.439 (0.004)       | **0.539 (0.004)** |           |
>
> **Notes**:
> - `++` indicates the row method is significantly better than the column method.
> - `--` indicates the row method is significantly worse than the column method.
>
> References:
>
>  1. James Bergstra, Rémi Bardenet, Yoshua Bengio, and Balázs Kégl. Algorithms for Hyper-Parameter Optimization. In Advances in Neural Information Processing Systems, pages 2546–2554, 2011.
>  2. Snoek, Jasper, Hugo Larochelle, and Ryan P. Adams. Practical bayesian optimization of machine learning algorithms. Advances in neural information processing systems, 25, 2012.

---

> ### Author Response · Authors · 2024-11-18
> **Reply to Reviewer XQkD [4]**
>
> > **What about the computational cost for IORBA vs. SBO and other mentioned baselines, what is this tradeoff?**
>
> Thank you for your question. Bayesian Optimization (BO) reduces computational costs by using surrogate models (e.g., Gaussian Processes) to predict hyper-parameter performance, significantly reducing the exhaustive trials required by methods like GridSearch and Randomized Optimization.
>
> The tradeoff in computational costs between IORBO and SBO is minimal. Specifically, IORBO incurs a slight additional cost for refitting the surrogate model with revised samples during the iterative refinement. However, this overhead is negligible compared to the overall computational cost. Apart from this refinement step, the process is essentially the same as SBO.
>
> We have added the following sentence to the section of IORBO:
>
>  > "IORBO incurs a slight additional cost for refitting the surrogate model with revised samples during the iterative refinement. However, this overhead is negligible compared to the overall computational cost. Apart from this refinement step, the process is essentially the same as SBO."
>
>
>
> > **Additionally, what are the optimized hyper-parameters that you obtain from your method? Ablation studies of the aforementioned would make your case stronger.**
>
> The optimized hyper-parameters obtained through IORBO vary depending on the dataset and GM used. In our experiments, we fine-tuned each GM on different datasets with two loss functions. The specific optimized hyper-parameters are included in our supplementary materials, but key examples include learning rates, batch sizes, epochs, network structures and the strength of regularization terms, all of which were tuned to maximize synthetic data quality across different evaluations. Please see the Appendix for more details.
>
>
> > **In TabSyn, the authors provided a comprehensive evaluation of synthetic tabular data using over five distinct evaluation metrics. Their metrics are straightforward and easy to comprehend. It will be nice to compare and justify why your metrics are more convincing and better than their proposed benchmark so that users should use your metrics instead of/in addition to TabSyn’s.**
>
> We appreciate the suggestion to compare our metrics with those used in TabSyn.
>
> In TabSyn, the authors employ five distinct metrics to assess synthetic data, each focusing on specific aspects of data quality. While these metrics are useful, they fall short of capturing the full range of complexities inherent in diverse datasets. Specifically, TabSyn evaluates only six datasets, which limits its ability to generalize across different domains.
>
> In contrast, our work aims to address this gap by incorporating a broader set of evaluation criteria. We evaluate synthetic data across a more diverse range of twenty datasets, enhancing the generalizability of our findings. Furthermore, our benchmarking framework provides a comprehensive evaluation by including (1) eight statistical metrics, (2) thirty metrics for regression tasks, and (3) sixty metrics for classification tasks. This diversity offers a more robust understanding of GM performance.
>
> While TabSyn's metrics can be easily integrated into our framework, our approach provides greater flexibility, as it allows for the addition of a wide variety of metrics. In contrast, expanding TabSyn's evaluation to the same scale would be more challenging.
>
> Our benchmarking framework allows users to assess model quality through **multiple complementary lenses** rather than a fixed set of metrics. This more inclusive benchmarking approach provides users with a robust toolkit for evaluating synthetic data quality, which we believe will inspire greater confidence in the reliability and versatility of GMs.
>
>
> > **Privacy is also crucial in synthetic tabular generation. How does your proposed loss function affect privacy-preserving metrics such as DCR and C2ST?**
>
> Thank you for highlighting this important point. Privacy is undoubtedly a critical factor in synthetic data generation, particularly when handling sensitive datasets. However, our current study focuses on publicly available datasets, where privacy concerns are less relevant. Consequently, our emphasis is on evaluating statistical fidelity and downstream ML performance, rather than on privacy-preserving metrics such as DCR and C2ST.

---

> ### Comment · Reviewer_XQkD · 2024-11-21
> **Response to Rebuttal**
>
> Thank you very much for the rebuttal.
>
> - I understand that the "three components you mentioned—loss function regularization, hyper-parameter tuning, and statistical tests—are interconnected". However, I believe that these components can also be introduced individually -- it is difficult to identify how each subsequent component reinforces the previous.
> - Thank you for the experiments to "evaluate IORBO against SBO using both mean and median aggregation methods". To further improve the paper, I feel it could still be beneficial to include other common practice benchmarks to justify your tuning method.
> - With regards to the benchmark, it is greatly appreciated that the authors have curated it. However, even though that TabSyn "evaluates only six datasets", I would appreciate it if the authors could justify why TabSYN "fall short of capturing the full range of complexities inherent in diverse datasets", specifically with regard to their metrics.
>
> I would be open to raising my score if my concerns can be addressed.

---

> > ### Author Response · Authors · 2024-11-25
> > **Reply to Reviewer XQkD [5]**
> >
> > > **I understand that the "three components you mentioned—loss function regularization, hyper-parameter tuning, and statistical tests—are interconnected". However, I believe that these components can also be introduced individually -- it is difficult to identify how each subsequent component reinforces the previous.**
> >
> > Thank you for the follow-up. The core of our approach remains the improvement of synthetic data quality, and the three components are designed to support this goal in a sequential, reinforcing manner. First, loss function regularization establishes a strong foundation by capturing data complexities. Second, hyper-parameter tuning (IORBO) builds on this by optimizing model performance, leveraging the loss function's potential. Finally, the statistical tests and benchmarking validate these improvements, ensuring practical impact. This pipeline ensures that each step enhances and supports the previous, forming a cohesive strategy.
> >
> >
> > > **Thank you for the experiments to "evaluate IORBO against SBO using both mean and median aggregation methods". To further improve the paper, I feel it could still be beneficial to include other common practice benchmarks to justify your tuning method.**
> >
> > We appreciate your suggestion to include common benchmarks like Grid Search and Randomized Search. However, our primary contribution lies in the ranking-based aggregation method (IOR), which is independent of the specific search method employed. While Grid Search and Randomized Search are viable search techniques, they could also benefit from our aggregation method, as it directly addresses the challenge of combining metrics with different scales or units in hyper-parameter optimization.
> >
> > By focusing on ranking metrics rather than traditional aggregation methods (e.g., mean or median), IOR avoids the pitfalls of unreliable aggregation, making it adaptable to search strategies, including Grid Search, Randomized Search, or Bayesian Optimization (BO). Our experiments intentionally compare IORBO against SBO with traditional mean and median aggregations to highlight the improvements our ranking-based method provides in making informed optimization decisions.
> >
> > We chose not to include Grid or Randomized Search comparisons because they are generally less efficient for hyper-parameter tuning in deep learning (Bergstra et al., 2011; Bergstra & Bengio, 2012). Including them would not enhance the evaluation of IOR, as our method addresses aggregation challenges rather than search methodologies. Instead, we demonstrate that IOR consistently improves performance across diverse objectives and aggregation methods.
> >
> > References:
> >
> >  1. Bergstra, James, and Yoshua Bengio. "Random Search for Hyper-Parameter Optimization." Journal of Machine Learning Research 13, no. 2 (2012): 281–305.
> >  2. Bergstra, James, Romain Bardenet, Yoshua Bengio, and Balázs Kégl. "Algorithms for Hyper-Parameter Optimization." In Advances in Neural Information Processing Systems, 2546–54. 2011.

---

> > > ### Author Response · Authors · 2024-11-25
> > > **Reply to Reviewer XQkD [6]**
> > >
> > > > **With regards to the benchmark, it is greatly appreciated that the authors have curated it. However, even though that TabSyn "evaluates only six datasets", I would appreciate it if the authors could justify why TabSYN "fall short of capturing the full range of complexities inherent in diverse datasets", specifically with regard to their metrics.**
> > >
> > > Thank you for your question. While TabSyn is a strong and valuable contribution to the field, there remain areas where further exploration and clarification could enhance its impact. Specifically, we see opportunities to address some knowledge gaps and extend the evaluation framework, particularly in model comparisons, dataset diversity, and evaluation metrics.
> > >
> > > **Model Comparison**
> > >
> > > TabSyn does not perform hyper-parameter tuning for the evaluated models, which leads to potentially unfair comparisons. Each generative model (GM) and dataset combination often requires specific hyper-parameters to achieve optimal performance. For example, in their paper, TabSyn reports that "TabDDPM fails to generate meaningful content on the News dataset." However, we observed that after performing hyper-parameter tuning for TabDDPM on the News dataset (which is one of our evaluated datasets), TabDDPM was able to generate meaningful and high-quality content. This finding suggests that TabSyn’s evaluation methodology could benefit from incorporating hyper-parameter tuning to ensure a fair comparison across all models.
> > >
> > > **Dataset Diversity**
> > >
> > > TabSyn evaluates only six datasets, all of which have fewer than 50,000 rows, representing medium-sized datasets. While this selection provides a good starting point, it remains unclear how TabSyn performs on smaller datasets (<2,000 rows) or larger datasets (>100,000 rows). Our work bridges this gap by including datasets across a wider range of sizes, ensuring the robustness of GMs in diverse scenarios.
> > >
> > > Additionally, TabSyn's datasets are all mixed-type, containing both categorical and continuous variables. This raises an open question: how do the models perform on datasets that are exclusively categorical or exclusively continuous? To address this, our benchmarks encompass datasets with varying data types to evaluate the adaptability of GMs across diverse tabular data structures.
> > >
> > > **Evaluation Metrics**
> > >
> > > TabSyn primarily evaluates generative models using five metrics:
> > >
> > >  - Alpha-precision
> > >  - Beta-recall
> > >  - MLE
> > >  - Pairwise Correlation
> > >  - Single Density
> > >
> > > Additionally, their supplementary materials (Section F.6) discuss the Distance to Closest Record (DCR) metric for differential privacy. However, the evaluation framework could benefit from greater clarity and broader metric diversity:
> > >
> > >  - Metric Scope: The chosen metrics primarily focus on fidelity, which is critical, but do not fully capture the full range of real-world requirements for synthetic data. For example, in evaluating Machine Learning Efficiency (MLE), TabSyn used XGBoost classifiers and regressors. But what if alternative ML methods, such as linear regression or random forests, were used to evaluate MLE? In such cases, if TabSyn underperforms compared to other models, can we still claim that it is the best-performing model overall? This suggests that a more diverse set of models and metrics would provide a more thorough evaluation of a model's generalizability and utility across different scenarios (Figueira et al., 2022).
> > >  - Use-Case Prioritization: The evaluation may not be entirely fair, as TabSyn did not perform hyper-parameter tuning for the tested models (as noted above). Assuming, however, that TabSyn outperforms other models on the five metrics, Table 13 (Section F.6) shows that STaSy exceeds TabSyn in terms of differential privacy (DCR). In scenarios where privacy is the primary concern—such as in healthcare or finance—STaSy would be considered the better model. This underscores the importance of aligning model evaluation with specific application priorities and use cases, rather than asserting a one-size-fits-all superiority (Raji et al., 2020).
> > >
> > >
> > > References:
> > >
> > >  1. Figueira, Alvaro, and Bruno Vaz. "Survey on synthetic data generation, evaluation methods and GANs." Mathematics 10, no. 15 (2022): 2733.
> > >  2. Raji, Inioluwa Deborah, Andrew Smart, Rebecca N. White, Margaret Mitchell, Timnit Gebru, Ben Hutchinson, Jamila Smith-Loud, Daniel Theron, and Parker Barnes. "Closing the AI accountability gap: Defining an end-to-end framework for internal algorithmic auditing." In Proceedings of the 2020 conference on fairness, accountability, and transparency, pp. 33-44. 2020.

---

> ### Comment · Reviewer_XQkD · 2024-11-25
> **Response to Rebuttal 2**
>
> **I understand that the "three components you mentioned—loss function regularization...**
>
> > Thank you for the follow-up. The core of our approach remains the improvement of synthetic data quality, and the three components are designed to support this goal in a sequential, reinforcing manner. First, loss function regularization establishes a strong foundation by capturing data complexities. Second, hyper-parameter tuning (IORBO) builds on this by optimizing model performance, leveraging the loss function's potential. Finally, the statistical tests and benchmarking validate these improvements, ensuring practical impact. This pipeline ensures that each step enhances and supports the previous, forming a cohesive strategy.
>
> Thanks for the reply. To clarify, the IORBO you introduced can be applied in scenarios **with or without** the loss function regularization or the statistical tests. Likewise for the loss function regularization as well as the statistical tests.
>
> **Thank you for the experiments to "evaluate IORBO against SBO using both mean and median aggregation methods". To further improve the paper, I feel it could still be beneficial to include other common practice benchmarks to justify your tuning method.**
>
> > We chose not to include Grid or Randomized Search comparisons because they are generally less efficient for hyper-parameter tuning in deep learning (Bergstra et al., 2011; Bergstra & Bengio, 2012). Including them would not enhance the evaluation of IOR, as our method addresses aggregation challenges rather than search methodologies. Instead, we demonstrate that IOR consistently improves performance across diverse objectives and aggregation methods.
>
> I understand that your method is superior and that the superiority is further emphasized here "(Bergstra et al., 2011; Bergstra & Bengio, 2012)". However, the mentioned fundamental baselines are important to further justify the superiority of your method and IOR over them. Just a simple comparison will do. This is analogous to TabSyn including SMOTE as a baseline.
>
> **With regards to the benchmark...**
>
> >  we observed that after performing hyper-parameter tuning for TabDDPM on the News dataset (which is one of our evaluated datasets), TabDDPM was able to generate meaningful and high-quality content.
>
> Would you happen to have ablations to justify the prominence of IORBA as the cause of improvement? (I am unable to find this in tables 3 and 5).
>
> > it remains unclear how TabSyn performs on smaller datasets (<2,000 rows) or larger datasets (>100,000 rows)
>
> Experiments to show that other metric baselines i.e. TabSyn will not work "on smaller datasets (<2,000 rows) or larger datasets (>100,000 rows)".
>
> I will definitely increase my score if my concerns can be addressed with experiments/ablation studies. My overall main concern still stands where there is no central theme of the paper.

---

> > ### Author Response · Authors · 2024-11-25
> > **Reply to Reviewer XQkD [7]**
> >
> > > **Thanks for the reply. To clarify, the IORBO you introduced can be applied in scenarios with or without the loss function regularization or the statistical tests. Likewise for the loss function regularization as well as the statistical tests.**
> >
> > Regarding your concern about the "central theme of the paper," we have revised the abstract and introduction (highlighted in blue in the updated manuscript) to clearly state that our primary goal is to improve synthetic tabular data quality. This involves addressing the complexities of real-world tabular data, such as diverse variable types, imbalances, and intricate dependencies, as well as overcoming challenges with aggregating metrics of different units. To achieve this, we have introduced three key components: IORBO, loss function regularization, and statistical tests, which collectively contribute to this objective.
> >
> > Could you please elaborate on what remains unclear or provide suggestions for improvement?
> >
> > We fully agree that IORBO can optimize other deep learning models, and the statistical tests can validate other methods. These features showcase the versatility of our contributions. However, in this work, we focus specifically on leveraging them to enhance synthetic tabular data generation.
> >
> > Could you please clarify your concern further or suggest specific improvements?
> >
> > P.S.: I will address your other comments shortly. Thank you again for your valuable feedback!

---

> ### Comment · Reviewer_XQkD · 2024-11-25
> **Reply**
>
> > Could you please elaborate on what remains unclear or provide suggestions for improvement?
> > Could you please clarify your concern further or suggest specific improvements?
>
> To be specific, I would focus the paper on either solely your proposed loss function or IORBO with more depth and detail but not both. I would like to reiterate again that the IORBO you introduced can be applied in scenarios with or without the loss function regularization or the statistical tests. Likewise, the loss function could also be applied in scenarios with or without the other two contributions. Hence, they don't actively reinforce/rely on one another to justify your contribution. Thus, there is no "central theme". The paper isn't coherent to a focused research question.

---

> > ### Author Response · Authors · 2024-11-27
> > **Reply to Reviewer XQkD [8]**
> >
> > > **I understand that your method is superior and that the superiority is further emphasized here "(Bergstra et al., 2011; Bergstra & Bengio, 2012)". However, the mentioned fundamental baselines are important to further justify the superiority of your method and IOR over them. Just a simple comparison will do. This is analogous to TabSyn including SMOTE as a baseline.**
> >
> > Thank you for your questions.
> >
> > In our study, we compared (1) the proposed loss function against the vanilla loss function and (2) IORBO against SBO with mean and median aggregation. These baselines-vanilla loss and SBO-were selected as they are appropriate to demonstrate the incremental contributions of our method. Our primary focus is to improve synthetic tabular data generation, not to benchmark generative models (GMs) comprehensively, as was the case in TabSyn. Including SMOTE, therefore, would not align with the specific objectives of our study.
> >
> > We hope this clarifies our approach and addresses your concern.
> >
> >
> > > **Experiments to show that other metric baselines i.e. TabSyn will not work "on smaller datasets (<2,000 rows) or larger datasets (>100,000 rows)".**
> >
> > To clarify, we did not state that TabSyn would not work on smaller datasets (<2,000 rows) or larger datasets (>100,000 rows). What we stated was the following:
> >
> >  > While this selection provides a good starting point, it remains unclear how TabSyn performs on smaller datasets (<2,000 rows) or larger datasets (>100,000 rows). Our work bridges this gap by including datasets across a wider range of sizes, ensuring the robustness of GMs in diverse scenarios.
> >
> > This statement emphasizes that the performance of TabSyn on datasets outside the range tested in its original paper has not been explored or demonstrated. Our intention was not to suggest that TabSyn would fail on such datasets, but rather to address the gap by evaluating GMs on datasets with more diverse sizes in our study.
> >
> >
> > > **To be specific, I would focus the paper on either solely your proposed loss function or IORBO with more depth and detail but not both. I would like to reiterate again that the IORBO you introduced can be applied in scenarios with or without the loss function regularization or the statistical tests. Likewise, the loss function could also be applied in scenarios with or without the other two contributions. Hence, they don't actively reinforce/rely on one another to justify your contribution. Thus, there is no "central theme". The paper isn't coherent to a focused research question.**
> >
> > Thank you for your suggestion. After extensive discussions among the authors, we agree with your point. In response, we have revised the Abstract and Introduction to better reflect the focus of the paper, as follows:
> >
> > **Abstract**:
> >
> >  > "To address these gaps, we introduce a novel correlation- and distribution-aware loss function that regularizes DGMs, enhancing their ability to generate synthetic tabular data that faithfully represents actual distributions. To aid in evaluating this loss function, we also propose a new multi-objective aggregation method using iterative objective refinement Bayesian optimization (IORBO) and a comprehensive statistical testing framework. While the focus of this paper is on improving the loss function, each contribution stands on its own and can be applied to other DGMs, applications, and hyperparameter optimization techniques."
> >
> > **Introduction**:
> >
> >  > "This work focuses on enhancing the performance of DGMs through a novel loss function, supported by a new multi-objective aggregation method and a comprehensive statistical testing framework that strengthen the performance and evaluation of our approach."

---

> > > ### Author Response · Authors · 2024-11-27
> > > **Reply to Reviewer XQkD [9]**
> > >
> > > > **Would you happen to have ablations to justify the prominence of IORBA as the cause of improvement? (I am unable to find this in tables 3 and 5).**
> > >
> > > Thank you for your question regarding the prominence of IORBO. In response, we have conducted additional ablation studies, and we have added the following sentence at the end of the Results section to highlight their importance:
> > >
> > >  > "Due to space limitations, further details on the ablation studies are presented in Section F. These studies emphasize the critical role of both the proposed loss function and IORBO optimization in enhancing model performance, with the combination of the two consistently yielding the best results across different configurations."
> > >
> > > You can now find Section F, which includes the ablation studies. In summary, these studies highlight the essential roles of both the proposed loss function and the IORBO optimization method in improving model performance. When combined, IORBO and the proposed loss function consistently delivers the best outcomes, validating the effectiveness of integrating both components for superior optimization performance.
> > >
> > > Additionally, please see the new Tables 9 and 10 below:
> > >
> > > **Table 9**: Results of the Nemenyi post-hoc test and win rate (with standard error in parentheses) comparing row and column methods. The table presents performance across different configurations, including the baseline with SBO and mean aggregation with the vanilla loss function, and comparisons with the proposed loss function and IORBO optimization method. For details on $p$-value ranges, refer to Table 1. "Van." and "Prop." denote the vanilla and proposed loss functions,
> > > respectively.
> > >
> > > |                        |            | Statistical Tests                      |          |          |          |            | Win Rate                               |          |          |          |
> > > |:------------------------:|:------------:|:----------------------------------------:|:----------:|:----------:|:----------:|:------------:|:----------------------------------------:|:----------:|:----------:|:----------:|
> > > | **Method**             |            | **SBO-Mean + Van.**                   | **IORBO + Van.** | **SBO-Mean + Prop.** | **IORBO + Prop.** |            | **SBO-Mean + Van.**                   | **IORBO + Van.** | **SBO-Mean + Prop.** | **IORBO + Prop.** |
> > > |                        |            |                                        |          |          |          |            |                                        |          |          |          |
> > > | **SBO-Mean + Van.**    |            |                                        | $--$     | $--$     | $--$     |            |                                        | 0.420 (0.004) | 0.419 (0.004) | 0.356 (0.003) |
> > > | **IORBO + Van.**         |            | $++$                                   |          | $++$     | $--$     |            | 0.580 (0.004)                          |            | 0.525 (0.004) | 0.418 (0.004) |
> > > | **SBO-Mean + Prop.**   |            | $++$                                   | $--$     |          | $--$     |            | 0.581 (0.004)                          | 0.475 (0.004) |            | 0.399 (0.004) |
> > > | **IORBO + Prop.**        |            | $++$                                   | $++$     | $++$     |          |            | 0.644 (0.003)                          | 0.582 (0.004) | 0.601 (0.004) |            |

---

> > > > ### Author Response · Authors · 2024-11-27
> > > > **Reply to Reviewer XQkD [10]**
> > > >
> > > > [continued from previous comment]
> > > >
> > > >
> > > > **Table 10**: Results of the Nemenyi post-hoc test and win rate (with standard error in parentheses) comparing row and column methods. The table presents performance across different configurations, including the baseline with SBO and median aggregation with the vanilla loss function, and comparisons with the proposed loss function and IORBO optimization method. For details on $p$-value ranges, refer to Table 1. "Van." and "Prop." denote the vanilla and proposed loss functions,
> > > > respectively.
> > > >
> > > > |                        |            | Statistical Tests                      |          |          |          |            | Win Rate                               |          |          |          |
> > > > |:------------------------:|:------------:|:----------------------------------------:|:----------:|:----------:|:----------:|:------------:|:----------------------------------------:|:----------:|:----------:|:----------:|
> > > > | **Method**             |            | **SBO-Med. + Van.**                   | **IORBO + Van.** | **SBO-Med. + Prop.** | **IORBO + Prop.** |            | **SBO-Med. + Van.**                   | **IORBO + Van.** | **SBO-Med. + Prop.** | **IORBO + Prop.** |
> > > > |                        |            |                                        |          |          |          |            |                                        |          |          |          |
> > > > | **SBO-Med. + Van.**    |            |                                        | $--$     | $--$     | $--$     |            |                                        | 0.454 (0.004) | 0.458 (0.004) | 0.400 (0.003) |
> > > > | **IORBO + Van.**         |            | $++$                                   |          | $0$      | $--$     |            | 0.546 (0.004)                          |            | 0.503 (0.004) | 0.418 (0.004) |
> > > > | **SBO-Med. + Prop.**   |            | $++$                                   | $0$      |          | $--$     |            | 0.542 (0.004)                          | 0.497 (0.004) |            | 0.423 (0.004) |
> > > > | **IORBO + Prop.**        |            | $++$                                   | $++$     | $++$     |          |            | 0.600 (0.004)                          | 0.582 (0.004) | 0.577 (0.004) |            |

---

> > > > > ### Comment · Reviewer_XQkD · 2024-11-27
> > > > > **Thank you for the experiments.**
> > > > >
> > > > > Thank you for the experiments. I have raised my score accordingly.

---

> > > > > > ### Author Response · Authors · 2024-12-02
> > > > > > **Thank you**
> > > > > >
> > > > > > Thank you for your prompt response.

---

### Official Review · Reviewer_1iCq · 2024-11-01

**Soundness:** 2
**Presentation:** 2
**Contribution:** 2
**Rating:** 5
**Confidence:** 3

**Summary:**

This paper introduces two regularization terms for improving the performance of the tabular generative model. The authors further propose to use ranking-based Bayesian Optimization to choose the hyperparameter. They finally evaluate the proposed method in Twenty tabular datasets on 10 base generative models by using TSTR, augmentation.

**Strengths:**

The experiments are comprehensive. Hyperparameters are chosen reasonably.

**Weaknesses:**

The proposed method is heuristic. The paper does not provide an optimality or convergence guarantee of the proposed loss. These two proposed losses are reasonable for tabular data but not general enough for other types of data. The hyperparameters are chosen by the new proposed Bayesian Optimization without theoretical guarantees.

**Questions:**

What will the performance if using Standard Bayesian optimization rather than IORBO proposed by this paper?

---

> ### Author Response · Authors · 2024-11-18
> **Reply to Reviewer 1iCq [1]**
>
> > **The proposed method is heuristic. The paper does not provide an optimality or convergence guarantee of the proposed loss.**
>
> Thank you for your questions. The primary aim of this work is to offer an effective approach to improve the quality of synthetic data specifically for real-world tabular datasets rather than achieving theoretical proofs.
>
> With careful fine-tuning, the empirical results across 20 datasets and 10 generative models (GMs) demonstrates that the proposed loss function consistently delivers statistically significant improvements in synthetic data quality and downstream machine learning (ML) performance. This fine-tuned empirical effectiveness indicates that the loss function is highly practical and impactful for the intended applications.
>
>
> > **These two proposed losses are reasonable for tabular data but not general enough for other types of data.**
>
> Our primary focus is to enhance the quality of synthetic tabular data, as clearly stated in the manuscript's title:
>
> > Improving Tabular Generative Models: Loss Functions, Benchmarks, and Iterative Objective Bayesian Approaches
>
> This focus is consistently emphasized throughout the abstract and the entire manuscript. We acknowledge that extending these methods to non-tabular data is a valuable avenue for future research, but it falls outside the scope of this work.

---

> ### Author Response · Authors · 2024-11-18
> **Reply to Reviewer 1iCq [2]**
>
> > **The hyper-parameters are chosen by the new proposed Bayesian Optimization without theoretical guarantees. What will the performance if using Standard Bayesian optimization rather than IORBO proposed by this paper?**
>
> While theoretical guarantees are of secondary importance, our primary objective is to achieve high performance with the proposed Iterative Objective Refinement Bayesian Optimization (IORBO). To address it, we conducted an additional experiment post-deadline to compare the IORBO with Standard Bayesian Optimization (SBO) using both mean and median aggregations. We fine-tuned each GM on a variety of datasets with different loss functions, applying each of the three Bayesian Optimization approaches.
>
> Statistical tests revealed that IORBO consistently outperformed SBO with both mean and median aggregations. As shown by the Nemenyi post-hoc test results and win rates (e.g., IORBO achieved win rates of 0.591 and 0.561 over SBO-Mean and SBO-Median, respectively), this significant performance gap demonstrates IORBO's robustness in handling multiple metrics across different units. It highlights that IORBO is effective for optimizing diverse objectives without requiring theoretical convergence guarantees.
>
> Additions/changes to the paper that we have made:
>
> 1. **Abstract:**
>    - *Previous Version*:
>      > "Further, the proposed IORBO outperformed the SBO with mean aggregation in terms of win rate and outperformed the SBO with median aggregation overall."
>    - *Updated Version*:
>      > "The IORBO consistently outperformed SBO, yielding superior optimization results."
>
> 2. **Section 3.6 - Benchmarking Framework:**
>    - *Added*:
>      > "Bayesian Optimization Method. To compare the performance of the IORBO with the SBO using mean and median aggregation methods, we fine-tuned each GM on each dataset across different loss functions, employing three evaluated BO approaches. Statistical tests were then conducted to evaluate these methods."
>
> 3. **Results Section:**
>    - *Previous Version*:
>      > "Bayesian optimization method. The performance of the IORBO was compared to the SBO using two aggregation methods (mean and median aggregation). The ML methods (Figure 1 and Step 2 and 3) were fine-tuned for each dataset using five-fold cross-validation on the ML evaluation metrics using different BO methods. The statistical tests were then employed to evaluate the three BO methods. Table 6 shows the results of the Nemenyi post-hoc test and win rate (with standard error in parentheses) comparing methods in the rows to those in the columns. The Nemenyi post-hoc test indicates that there is no significant difference between IORBO and SBO-Mean, but in terms of the win rate, the IORBO performs significantly better than the SBO-Mean with a win rate of 0.527. The IORBO method demonstrates significant improvement compared to the SBO-Median method, both in terms of the Nemenyi post-hoc test ($++$) and in terms of the win rate (0.534)."
>
>    - *Updated Version*:
>      > "Bayesian Optimization Method. The performance of the IORBO was compared to the SBO using two aggregation methods (mean and median aggregation). We fine-tuned each GM on each dataset across two loss functions, employing three evaluated BO approaches. Table 6 shows the results of the Nemenyi post-hoc test and win rate (with standard error in parentheses) comparing methods in the rows to those in the columns. The Nemenyi post-hoc test indicates that the IORBO is significantly better than the SBO-Mean and SBO-Median with win rates of 0.591 and 0.561, respectively. The results demonstrate that IORBO is robust in handling metrics with different units and its potential as a reliable, broadly applicable BO method."

---

> ### Author Response · Authors · 2024-11-18
> **Reply to Reviewer 1iCq [3]**
>
> [continued from previous comment]
>
> 4. **Table 5**
>    - *Previous Version*:
>      > Table 5: Results of the Nemenyi post-hoc test and win rate (with standard error in parentheses) comparing the row to column method. For details on significance levels, refer to Table 1.
>
> | **BO Method**        |             | **IOR**            | **SBO-Mean**       | **SBO-Median**      |             | **IOR**            | **SBO-Mean**       | **SBO-Median**      |
> |----------------------|-------------|--------------------|--------------------|---------------------|-------------|--------------------|--------------------|---------------------|
> |                      |             | **Statistical Tests** |                    |                     |             | **Win Rate**       |                    |                     |
> |                      |             | IOR                | SBO-Mean           | SBO-Median          |             | IOR                | SBO-Mean           | SBO-Median          |
> | **IOR**              |             |                    | 0                  | ++                  |             | **0.527 (0.010)**   | **0.534 (0.010)**   |                     |
> | **SBO-Mean**         |             | 0                  |                    | ++                  |             | 0.473 (0.010)      |                    | **0.543 (0.010)**   |
> | **SBO-Median**       |             | --                 | --                 |                     |             | 0.466 (0.010)      | 0.457 (0.010)      |                     |
>
>    - *Updated Version*:
>      > Table 5: Results of the Nemenyi post-hoc test and win rate (with standard error in parentheses) comparing the row to column method. For details on significance levels, refer to Table 1.
>
> | **BO Method**         |             | **Statistical Tests** |           |           |             | **Win Rate**        |           |           |
> |-----------------------|-------------|-----------------------|-----------|-----------|-------------|---------------------|-----------|-----------|
> |                       |             | **IOR**              | **SBO-Mean** | **SBO-Median** |             | **IOR**            | **SBO-Mean** | **SBO-Median** |
> |                       |             |                       |           |           |             |                     |           |           |
> | **IOR**               |             |                       | ++        | ++        |             | **0.591 (0.004)**   | **0.561 (0.004)** |           |
> | **SBO-Mean**          |             | --                    |           | --        |             | 0.409 (0.004)       |           | 0.461 (0.004) |
> | **SBO-Median**        |             | --                    | ++        |           |             | 0.439 (0.004)       | **0.539 (0.004)** |           |
>
> **Notes**:
> - `++` indicates the row method is significantly better than the column method.
> - `--` indicates the row method is significantly worse than the column method.

---

> > ### Comment · Reviewer_1iCq · 2024-12-01
> >
> > Thank the authors for the response. I have carefully read the rebuttal. I understand the claim of the authors. I will keep the score.

---

> > > ### Author Response · Authors · 2024-12-02
> > > **Thank you**
> > >
> > > Thank you for your prompt response.

---

### Author Response · Authors · 2024-11-18
**To all reviewers**

Thank you to all reviewers for your valuable comments. We greatly appreciate your feedback. A revised version of our manuscript has been uploaded, with all changes and updates highlighted in blue. We look forward to your response.

---

### Comment · Area_Chair_fGk5 · 2024-11-25
**Discussions between reviewers and authors**

Time for discussions as author feedback is in. I encourage all the reviewers to reply. You should treat the paper that you're reviewing in the same way as you'd like your submission to be treated :)

---

### Author Response · Authors · 2024-12-02
**To all Reviewers**

We would like to express our gratitude to all the reviewers for their constructive feedback and insightful comments. Below, we provide a summary of the key changes and additions made in response to your suggestions.


---

**Reviewer 1iCq:**

- *Query on Standard Bayesian Optimization (SBO):* To address your query, we conducted additional experiments comparing our proposed IORBO with SBO. The results, included in the revised manuscript, demonstrate that IORBO consistently outperforms SBO.

---

**Reviewer XQkD:**

- *Central Theme and Research Flow:* We acknowledge your concern regarding the interconnectedness of our three contributions (loss function regularization, hyperparameter optimization via IORBO, and statistical tests). In response, we have clarified in both the abstract and introduction that these components can function independently or as a unified framework, with each addressing specific challenges in synthetic tabular data generation.
- *Comparison with TabSyn Datasets, Metrics, and Models:*   While TabSyn is an important contribution to the field, we have highlighted the gaps our method addresses in benchmarking metrics and its broader applicability.
- *Privacy Metrics:*   While privacy-preserving metrics (e.g., DCR, C2ST) are critical in synthetic data research, our study focuses on publicly available datasets where privacy concerns are less relevant. As such, we emphasize evaluating statistical fidelity and downstream ML performance rather than privacy-specific metrics.
- *Ablation Studies:* Additional ablation studies were conducted to evaluate the contributions of the loss function and IORBO. Included in the supplementary material, these studies highlight the significant impact of both components on model performance. While each contributes positively, the combination consistently delivers the best results, emphasizing their complementary roles in optimization.



---

**Reviewer mC61:**

- *Biased Estimator:* Regarding the use of biased estimators for standard deviation (and moments), we have clarified that mini-batch training is essential for efficient deep learning. While it introduces bias, this trade-off allows for handling large datasets, reducing memory usage, and enabling frequent model updates.
- *Higher-Order Moments:*   We acknowledge the potential numerical instability when incorporating higher-order moments. As a solution, we propose using an exponential moving average of moments over iterations to ensure stability. This suggestion has been added as a potential future improvement in the conclusion.
- *Validation of Lower-Order Moments:*   Our approach is rooted in the method of moments, a well-established statistical technique, we have offered some intuitive reasoning: The loss can be thought of as "pinning down" the low-frequency components of the distribution (i.e., the lower-order moments), which stabilizes the GM's learning process by ensuring that it does not deviate significantly from the observed distribution in these aspects.
- *Multimodal Distributions:* The distribution-aware loss doesn’t need to fully represent the entire data distribution. Instead, the generative model learns additional characteristics, like the positions and weights of modes in multimodal distributions. Empirical results show that aligning lower-order moments (up to the 4th) captures key characteristics of tabular data.

---

**Reviewer HdF4:**

- *Data Heterogeneity and Missing Data:* We have expanded the discussion on handling data heterogeneity and missing data. This includes addressing challenges such as the dominance of one data type over others and proposing strategies to mitigate this risk. Additionally, we emphasize how our approach balances regularization terms across diverse variable types.
- *Evaluation of Metrics by Data Type:*   While we currently apply a unified distribution-aware loss function to all data types (e.g., continuous, non-ordered discrete, ordinal, and Poisson-like variables), our results in Tables 2–4 demonstrate significant improvements in fidelity and downstream ML performance. This indicates the effectiveness of our general approach despite the lack of separate regularizers for different data types.
- *Behavior of Optimization Across Variable Types:*  We analyzed the impact of variable types (discrete vs. continuous) on optimization, considering factors such as: the ratio of discrete to continuous variables, normalization methods, and encoding schemes.

---

Once again, we deeply appreciate the time and effort invested by the reviewers in providing such detailed feedback. We believe these revisions have significantly strengthened the manuscript and addressed the key concerns raised. Thank you!

---

### Meta-Review · Area_Chair_fGk5 · 2024-12-21

**Metareview:**

This paper introduces a correlation- and distribution-aware loss function for tabular data generative model training. In addition, the authors improved Bayesian optimisation for tuning the hyper-parameters of the model and training procedure. They also introduced an evaluation benchmark to compare the performance for various number of tabular generative models.

Reviewers have various concerns in their initial reviews, regarding both the technical details and the confusion on the central contribution of the paper. Author rebuttal addressed some, but not all, of the concerns, see below "additional comments on reviewer discussion".

After a brief read, I also have the feeling that, 3 rather independent ideas are presented in the paper. For the significance of the contribution, I checked whether

(1) the paper is stellar in arguing each of the 3 independent ideas thoroughly; or

(2) the paper combines the 3 ideas in a clever way to make an overall significant contribution.

Unfortunately the answer for both options are NO.

- First the benchmarking framework needs to be justified independently -- why this evaluation method is better than some other frameworks?

- Second, if the authors were to claim that the new training loss + new BO procedure give the best generative model, then they need to provide ablation study regarding both components. E.g., would training the model without the new training loss + standard BO perform similarly?

I encourage the authors to revise their manuscript by incorporating the suggestions from this reviewing process.

**Additional Comments On Reviewer Discussion:**

In AC-reviewer discussions, further comments were provided, see below.

Comment 1

"
The author has yet to present any rigorous proof or detailed justification in the revised manuscript about the consistency/correctness of the proposed L_distribution. While there is an intuition-level explanation in the discussion on OpenReview (which was not incorporated into the manuscript), there is no rigorous mathematical proof or specific experiment on this for readers to fully endorse it. This omission impacts the credibility of the proposed method, just like other reviewer's concern that the proposed losses appear to be "heuristic".

Particularly for the experiment, there are no ablation experiments isolating the effects of L_distribution or L_correlation alone. The current results (e.g., Tables 9 and 10) focus only on the combined effect of the overall proposed loss but fail to clarify the individual contributions of these components. This is particularly problematic given that L_distribution​ lacks a rigorous foundation, which raises questions about its reliability and utility for the machine learning community.

The dearth of these analyses limits the paper's contribution. I believe additional work is needed to strengthen the theoretical and/or experimental grounding of the submission towards publication. And, my rating remains unchanged.
"

Comment 2

"
On my side I would like to have seen more empirical evidence on how the method behaves in heterogeneus data, particularly if training favours some data types rather than others or how the author can compensate for that.
"

---

### Decision · Program_Chairs · 2025-01-22

Reject